# Chromosome-level genome assemblies of 2 hemichordates provide new insights into deuterostome origin and chromosome evolution

Che-Yi Lin[1], Ferdinand Marlétaz[2], Alberto Pérez-Posada[3,4], Pedro Manuel Martínez-García[3], Siegfried Schloissnig[5], Paul Peluso[6], Greg T. Conception[6], Paul Bump[7], Yi-Chih Chen[1], Cindy Chou[1], Ching-Yi Lin[1], Tzu-Pei Fan[1], Chang-Tai Tsai[1], José Luis Gómez Skarmeta[3†], Juan J. Tena[3], Christopher J. Lowe[7,8], David R. Rank[6], Daniel S. Rokhsar[8,9,10]*, Jr-Kai Yu [1,11]*, Yi-Hsien Su[1]*

1 Institute of Cellular and Organismic Biology, Academia Sinica, Taipei, Taiwan, 2 Center for Life's Origins and Evolution, Department of Genetics, Evolution and Environment, University College London, London, United Kingdom, 3 Centro Andaluz de Biología del Desarrollo, Consejo Superior de Investigaciones Científicas-Universidad Pablo de Olavide-Junta de Andalucía, Seville, Spain, 4 Living Systems Institute, University of Exeter, Exeter, United Kingdom, 5 Vienna Biocenter, Institute of Molecular Pathology, Vienna, Austria, 6 Pacific Biosciences, Menlo Park, California, United States of America, 7 Hopkins Marine Station, Department of Biology, Stanford University, Pacific Grove, California, United States of America, 8 Chan-Zuckerberg Biohub, San Francisco, California, United States of America, 9 Department of Molecular and Cell Biology, University of California, Berkeley, Berkeley, California, United States of America, 10 Molecular Genetics Unit, Okinawa Institute for Science and Technology, Onna, Japan, 11 Marine Research Station, Institute of Cellular and Organismic Biology, Academia Sinica, Yilan, Taiwan

† Deceased.
* dsrokhsar@gmail.com (DSR); jkyu@gate.sinica.edu.tw (J-KY); yhsu@gate.sinica.edu.tw (YHS)

**Data Availability Statement:** P. flava genome assembly used in this work is publicly available: https://www.ncbi.nlm.nih.gov/bioproject/

## Abstract

Deuterostomes are a monophyletic group of animals that includes Hemichordata, Echino-dermata (together called Ambulacraria), and Chordata. The diversity of deuterostome body plans has made it challenging to reconstruct their ancestral condition and to decipher the genetic changes that drove the diversification of deuterostome lineages. Here, we generate chromosome-level genome assemblies of 2 hemichordate species, *Ptychodera flava* and *Schizocardium californicum*, and use comparative genomic approaches to infer the chromosomal architecture of the deuterostome common ancestor and delineate lineage-specific chromosomal modifications. We show that hemichordate chromosomes ($1N = 23$) exhibit remarkable chromosome-scale macrosynteny when compared to other deuterostomes and can be derived from 24 deuterostome ancestral linkage groups (ALGs). These deutero-stome ALGs in turn match previously inferred bilaterian ALGs, consistent with a relatively short transition from the last common bilaterian ancestor to the origin of deuterostomes. Based on this deuterostome ALG complement, we deduced chromosomal rearrangement events that occurred in different lineages. For example, a fusion-with-mixing event produced an Ambulacraria-specific ALG that subsequently split into 2 chromosomes in extant hemi-chordates, while this homologous ALG further fused with another chromosome in sea urchins. Orthologous genes distributed in these rearranged chromosomes are enriched for functions in various developmental processes. We found that the deeply conserved Hox

PRJNA747109. The version described in this paper is version JASXRY010000000 (https://submit.ncbi.nlm.nih.gov/api/2.0/files/z1apzwkx/po1410_ptychodera_flava.repeatmasked.fasta/?format=attachment). Genome assembly and gene annotation files can be downloaded from https://figshare.com/projects/Hemichordate_Genomes/168110.

**Funding:** This work was supported by grants 112-2326-B-001-004 (Y.H.S.) and 110-2621-B-001-001-MY3 (J.K.Y.) from the National Science and Technology Council, Taiwan (https://www.nstc.gov.tw/?l=en), grant AS-GC-111-L01 from Academia Sinica, Taiwan (https://www.sinica.edu.tw/en/) (Y.H.S and J.K.Y.), and grant PID2019-10921GB-I00 from Ministerio de Economía y Competitividad, Spain (https://portal.mineco.gob.es/en-us/Pages/index.aspx) (J.J.T.). P.M.M.G. was funded by a postdoctoral fellowship from Junta de Andalucía (https://www.juntadeandalucia.es/) (DOC_0097). F.M. is supported by the Royal Society Fellowship (https://royalsociety.org/) URF \R1\191161 and the BBSRC grant BB/V01109X/1 (https://www.ukri.org/councils/bbsrc/). D.S.R. was supported by the Molecular Genetics Unit at the Okinawa Institute for Science and Technology (https://www.oist.jp/), and is grateful for support from the Marthella Foskett Brown Chair in Biological Sciences at UC Berkeley (https://www.berkeley.edu/). D.S.R. and C.J.L. were supported by the Chan Zuckerberg BioHub (https://www.czbiohub.org/). The sponsors or funders play no role in the study design, data collection and analysis, decision to publish, or preparation of the manuscript.

**Competing interests:** D.S.R. is the paid consultant and shareholder of Dovetail Genomics. The other authors have declared that no competing interests exist.

**Abbreviations:** AAG, *Aricia agestis*; ALG, ancestral linkage group; AMI, *Acropora millepora*; APL, *Acanthaster planci*; AQU, *Amphimedon queenslandica*; BALG, bilaterian ancestral linkage group; BFL, *Branchiostoma floridae*; CALG, chordate ALG; CHE, *Clytia hemisphaerica*; CGI, *Crassostrea gigas*; CRO, *Carcinoscorpius rotundicauda*; EAE, *Erebia aethiops*; EMU, *Ephydatia muelleri*; GO, gene ontology; HSA, *Homo sapiens*; HGL, *Heterodera glycines*; HMW, high molecular weight; LCA, last common ancestor; LINE, long interspersed nuclear elements; LPI, *Lytechinus pictus*; LTR, long terminal repeats; LVA, *Lytechinus variegatus*; mya, million years ago; NVE, *Nematostella vectensis*; PCH, *Penaeus chinensis*; PEC, *Paraescarpia echinospica*; PFL, *Ptychodera flava*; PMI, *Patiria miniata*; POC,

clusters are located in highly rearranged chromosomes and that maintenance of the clusters are likely due to lower densities of transposable elements within the clusters. We also provide evidence that the deuterostome-specific pharyngeal gene cluster was established via the combination of 3 pre-assembled microsyntenic blocks. We suggest that since chromosomal rearrangement events and formation of new gene clusters may change the regulatory controls of developmental genes, these events may have contributed to the evolution of diverse body plans among deuterostomes.

## Introduction

The evolutionary events that gave rise to the diverse body plans of deuterostomes remains one of the major mysteries in biology. It is widely accepted that the Deuterostomia includes Echinodermata, Hemichordata, and Chordata, as these animals are characterized by several unique developmental and morphological features, including radial cleavage, deuterostomy, enterocoely formation of the mesoderm, mesoderm-derived skeletal tissues, and pharyngeal openings/slits [1–3]. Despite these common characters, the different deuterostome lineages have evolved distinct body plans. Chordates are defined by their dorsal tubular central nervous system, notochord, and segmented somites [4], while echinoderms evolved a pentaradially symmetrical adult body, calcitic endoskeleton, and a water vascular system [5]; and hemichordates are characterized by a tripartite body organization, which includes a proboscis, collar, and trunk [6]. Molecular phylogenetic analyses have supported a sister group relationship between Echinodermata and Hemichordata, forming a clade called Ambulacraria [3,7,8] (Fig 1A). While subsequent phylogenomic studies have reinforced support for the ambulacrarian clade, some have suggested a sister group relationship between Ambulacraria and Xenacoelomorpha (a group of marine worms lacking definitive coeloms) and even questioned the monophyletic grouping of the Deuterostomia [9–12]. Due to the long evolutionary history of deuterostome lineages and the difficulties in assigning definitive stem fossils during the early diversification of the group, it remains challenging to postulate the ancestral condition of their common ancestor, let alone to decipher the genomic basis underlying the origins of diverse body plans and phylogenetic affiliations. To address these issues, it is helpful to reconstruct the ancestral genome architectures at major nodes of the animal tree using species that occupy key phylogenetic positions, and trace the subsequent evolutionary trajectories along each lineage.

Comparison of diverse metazoan genomes has revealed extensive conservation of chromosome-scale linkage (i.e., "macrosynteny") across animals [13–17] and enabled the reconstruction of ancestral chromosome-scale units (chromosomes or chromosome arms) [18–21]. These reconstructions have been used to identify shared and derived synteny patterns that can help to resolve long-standing evolutionary questions, infer lineage-specific chromosomal rearrangements, and clarify animal phylogenetic relationships that have been difficult to resolve using conventional phylogenetic approaches [18–23]. For example, identifications of synapomorphic traits of chromosomal fusion-with-mixing events among sponge, cnidarian, and bilaterian genomes provide strong evidence to support the hypothesis that ctenophores are the sister group to all other animals [18].

Among deuterostomes, vertebrates show extensive genomic duplications [20], but comparisons of sea urchin with other bilaterians [19], and analysis of sub-chromosomal assemblies of hemichordates [15] (1) implied that the chromosomes of the deuterostome ancestor retained the 24 bilaterian ancestral linkage groups (BALGs); and (2) identified subsequent

*Pisaster ochraceus*; PYE, *Patinopecten yessoensis*; RES, *Rhopilema esculentum*; RPH, *Ruditapes philippinarum*; SBE, *Streblospio benedicti*, SCA, *Schizocardium californicum*; SCAL, *Scolanthus callimorphus*; SCO, *Sinonovacula constricta*; SINE, short interspersed nuclear elements; SMA, *Sanderia malayensis*; SPU, *Strongylocentrotus purpuratus*; TAD, topological association domain; TE, transposable element; TTR, *Tachypleus tridentatus*; XSP, *Xenia* sp.

rearrangement in the sea urchin and chordate lineages [19]. Assembling a complete picture of deuterostome genome evolution, however, requires comparisons including chromosome-scale assemblies of hemichordates. Analyses of karyotype evolution including all deuterostome phylum-level lineages could yield important insights into deuterostome ancestry and the evolution of their diverse body plans.

Hemichordates comprise 2 groups, the solitary enteropneusts and the colonial pterobranchs. In this study, we generated chromosome-level genome assemblies for 2 enteropneusts, the ptychoderid *Ptychodera flava* and spengelid *Schizocardium californicum*. Phylogenomic data showed that Ptychoderidae and Spengelidae are sister groups, together with Harrimaniidae constituting Enteropneusta [7,24]. Our comparative genomic analysis showed remarkable macro-syntenic conservation among deuterostome species. Based on the principle of parsimony and comparative analyses with outgroups, we deduced that the last common ancestor of deuterostomes possessed 24 ancestral linkage group (ALGs) that match the BALGs as previously proposed [19]. We also discovered lineage-specific rearrangements that reflect the temporal progression towards the chromosomal architectures of extant deuterostomes. While our phylogenetic analysis using synteny-based characters supports a monophyletic deuterostome grouping, we did not identify shared derived macrochromosomal rearrangements that distinguish deuterostomes from other bilaterians. Our results confirm that the genomic architectures of deuterostomes retain more ancestral traits than those of protostomes, consistent with a very short evolutionary distance from the last common ancestor of bilaterians to the origin of deuterostomes. Our study thus provides a roadmap for understanding chromosomal evolution and contributes to deciphering the possible developmental genetic changes underlying the emergence of diverse body plans in deuterostomes.

## Results and discussion

### Chromosome-level genome assemblies of 2 hemichordates

Deuterostomes are composed of 3 major phyla, including hemichordates, echinoderms, and chordates, with the former 2 constituting a group called Ambulacraria (Fig 1A). Previous short

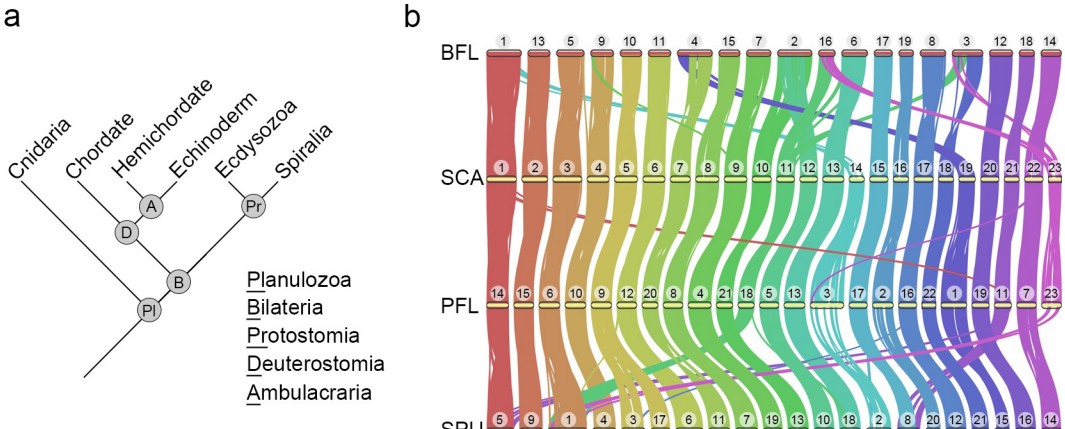

**Fig 1. Highly conserved macrosyntenic structure among deuterostomes was detected based on chromosome-level genome assemblies, including 2 new hemichordate genomes.** (**a**) A simplified phylogenetic tree of major branches in Planulozoa (Cnidaria+Bilateria). (**b**) Macrosynteny conservation among deuterostome species including BFL, SCA, PFL, and SPU. Horizontal bars with numbers above represent chromosomes of each species. The conserved synteny blocks between 2 species are connected by curve lines (minimum of 4 gene pairs within a maximum distance of 75 genes between 2 matches). The data underlying this figure can be found in S1 Data. BFL, *Branchiostoma floridae*; PFL, *Ptychodera flava*; SCA, *Schizocardium californicum*; SPU, *Strongylocentrotus purpuratus*.

read-based genome sequencing of 2 hemichordate species, *Saccoglossus kowalevskii* and *Ptychodera flava*, provided a cornerstone for studies on deuterostome evolution [15]. The fragmented nature of these genome assemblies, however, limits our understanding of chromosome evolution among deuterostome lineages. To address this issue, we employed PacBio long-read and HiC technologies to sequence genomes of 2 enteropneust hemichordates *P. flava* (PFL) and *Schizocardium californicum* (SCA) (S1 Fig). The long read-based genome assemblies of PFL and SCA consist of 1.16 Gbp and 0.93 Gbp, respectively (S1 Fig). After consideration of HiC contacts (S2 Fig), 23 chromosome-scale scaffolds were obtained for both genomes, which matches the $2N = 46$ karyotype of PFL [15]. Protein-coding genes were annotated in the 2 genome assemblies using transcriptome data and ab initio prediction approaches, resulting in 35,856 (PFL) and 27,463 (SCA) annotated genes with high BUSCO scores (S1 Fig). Therefore, these 2 hemichordate genome assemblies reached chromosome level with high completeness in gene annotation.

The 23 chromosomes of the 2 hemichordate species generally exhibit a one-to-one correspondence based on pairwise comparisons of the positions of orthologous genes (Figs 1B and S3A). This correspondence further supports the chromosomal-scale accuracy of the independently conducted genome assemblies, since conserved syntnies are unlikely to be generated spuriously by assembly errors. Extending this analysis to sea urchin (*Strongylocentrotus purpuratus*, SPU) and amphioxus (*Branchiostoma floridae*, BFL), which are representative echinoderm and chordate species, we confirmed chromosome-scale syntenic conservation (macrosynteny) among deuterostomes (Figs 1B and S3B). Given that macro-syntenic conservation has been used to reconstruct ancestral genome architectures and identify lineage-specific chromosomal rearrangement events [19,20], we broadened the synteny analysis by including additional species within and outside the deuterostome superphylum. This approach allowed us to confirm the genomic architecture of the last common ancestor (LCA) of deuterostomes and explore how it evolved among deuterostome lineages.

## Reduction of chromosome numbers during deuterostome evolution

To reconstruct the ancestral chromosomal architectures at key phylogenetic nodes in deuterostomes and investigate the evolutionary history of chromosomal changes, we carried out pairwise genome comparisons of multiple deuterostomes (S4 Fig). To identify orthologous chromosomes between species in an unbiased fashion, we employed Fisher's exact test with Bonferroni correction and risk difference to designate chromosome pairs containing orthologous genes (see Methods). Following refs. 19 and 20, we reasoned that the syntenic units that are conserved between genomes are most likely descended from a common ALG in the LCA of the 2 species under investigation. We used the scallop (*Patinopecten yessoensis*, PYE) genome as an outgroup (S5 Fig) due to its slow evolution compared with other protostomes [25] and previously demonstrated conserved syntenies with other animals [19]. Using this comparative approach, we inferred ancestral chromosomal architectures at major nodes of the deuterostome phylogeny.

In order to reconstruct the ambulacrarian ancestral chromosomes, we compared the hemichordate PFL genome with the genomes of 2 echinoderm species (sea urchin SPU and sea star *Pisaster ochraceus*, POC), with the amphioxus or scallop genome serving as an outgroup (S6–S9 Figs). The dot plot between hemichordate (PFL) and sea urchin (SPU) showed 17 one-to-one corresponding chromosomes (S4A Fig), suggesting that (1) these chromosome pairs are homologous; and (2) the LCA of PFL and SPU (i.e., the ambulacrarian LCA) already possessed these 17 ALGs. We also identified several one-to-two and one-to-three corresponding chromosomes between PFL and SPU, implying that large-scale chromosomal rearrangement

events occurred after the lineages diverged from the ambulacrarian LCA. We polarized the direction of chromosomal change and identified the likely ancestral state by comparing to the outgroup species. For example, *P. flava* PFL11 and PFL17 together correspond to *S. purpuratus* SPU8 (S8D Fig), implying that either PFL11 and PFL17 arose by a split of an ancestral ambulacrarian chromosome or SPU8 arose by the fusion of 2 ancestral chromosomes. Comparison with amphioxus chromosomes, however, showed that PFL11 and PFL17 respectively correspond to amphioxus BFL18 and BFL17 (S8G Fig), indicating that these 2 chromosome pairs evolved from 2 distinct ALGs in the deuterostome LCA. Based on the parsimony principle, we reasoned that hemichordates inherited the 2 ALGs directly as PFL11 and PFL17, while sea urchin SPU8 was fused from the 2 distinct ancestral chromosomes, as also noted in ref. 19 using a different sea urchin species *Lytechinus variegatus* (S8A Fig). By reiterating such comparisons (S6–S9 Figs), we find that the LCA of deuterostomes possessed 24 ALGs (DALGs). Importantly, these 24 DALGs correspond to the 24 BALGs deduced by Simakov and colleagues [19], confirming that the deuterostome LCA and the bilaterian LCA possessed very similar chromosomal architectures. Our notation for the deuterostome ALGs therefore follows those of the bilaterian ALGs [19]. Among the 24 DALGs, 9 remain intact in all 5 deuterostome species we investigated, while 15 have undergone lineage-specific changes (Fig 2).

Fig 2 illustrates chromosomal rearrangement events with color boxes: interspersed boxes represent chromosomal fusions followed by translocations, while checkerboards depict chromosomal fusions followed by extensive mixing, which is a common feature of deep chromosome evolution [19] (Fig 2); rearrangements were determined based on pairwise conserved syntenies between target species (S6–S9 Figs). These illustrations correspond to the chromosomal rearrangement events defined by Simakov and colleagues [19], with algebraic symbols indicating end-end fusion ($\bullet$), centric insertion ($\searrow$), and fusion-with-mixing ($\otimes$) [19]. Notably, 4 interspersed boxes correspond to end-end fusions and 5 correspond to centric insertions followed by chromosomal translocations (e.g., BFL4 and BFL2 in Fig 2B). From the 24 DALGs, we inferred that the numbers of chromosomes were reduced in a lineage-specific manner. In the lineage leading to ambulacrarians (node A in Fig 2B), DALGs B2 and C2 fused and mixed extensively to become the ambulacrarian ALG B2$\otimes$C2, while other DALGs remained relatively intact, resulting in 23 ambulacrarian ALGs (AALGs). In the hemichordate lineage (node H in Fig 2B), AALG B2$\otimes$C2 split into 2 chromosomes (B2$\otimes$C2-a and B2$\otimes$C2-b), while AALGs R and B1 fused and mixed (AALG R$\otimes$B1) to become a single chromosome (PFL9 and SCA5, respectively), resulting in $1N = 23$ chromosomes in both hemichordate species. The split of the AALG B2$\otimes$C2 can be understood as a possible Robertsonian (i.e., centric) fission in which a presumably metacentric chromosome is transformed into 2 acrocentrics. Whether the shared chromosomal linkages of PFL and SCA represents the ancestral hemichordate state can only be determined by analysis of pterobranch hemichordate genomes, but it is clear from the pairwise comparison (S3A Fig) that no large-scale macrosyntenic changes have occurred since the last common ancestor of PFL and SCA, which lived more than 370 mya [15].

Similarly, the echinoderm LCA (node E in Fig 2B) likely possessed 23 ALGs (EALGs), with the same chromosomal architecture as the ambulacrarian LCA; subsequently, different fusion events occurred in the sea star and sea urchin lineages. In the sea star, EALGs O2 and B3 fused (O2$\otimes$B3) and evolved into POC6, resulting in a $1N = 22$ karyotype. In the sea urchin *S. purpuratus*, SPU8 chromosome arose through the fusion of EALGs J1 and B3, via central insertion (J1$\searrow$B3), while SPU1 arose by fusion with extensive mixing from EALGs E and B2$\otimes$C2, denoted as E$\otimes$(B2$\otimes$C2), resulting in $1N = 21$ chromosomes. The three-way fusion E$\otimes$(B2$\otimes$C2), is also shared by *Lytechinus variegatus* [19] and *Paracentrotus lividus* [26], and is

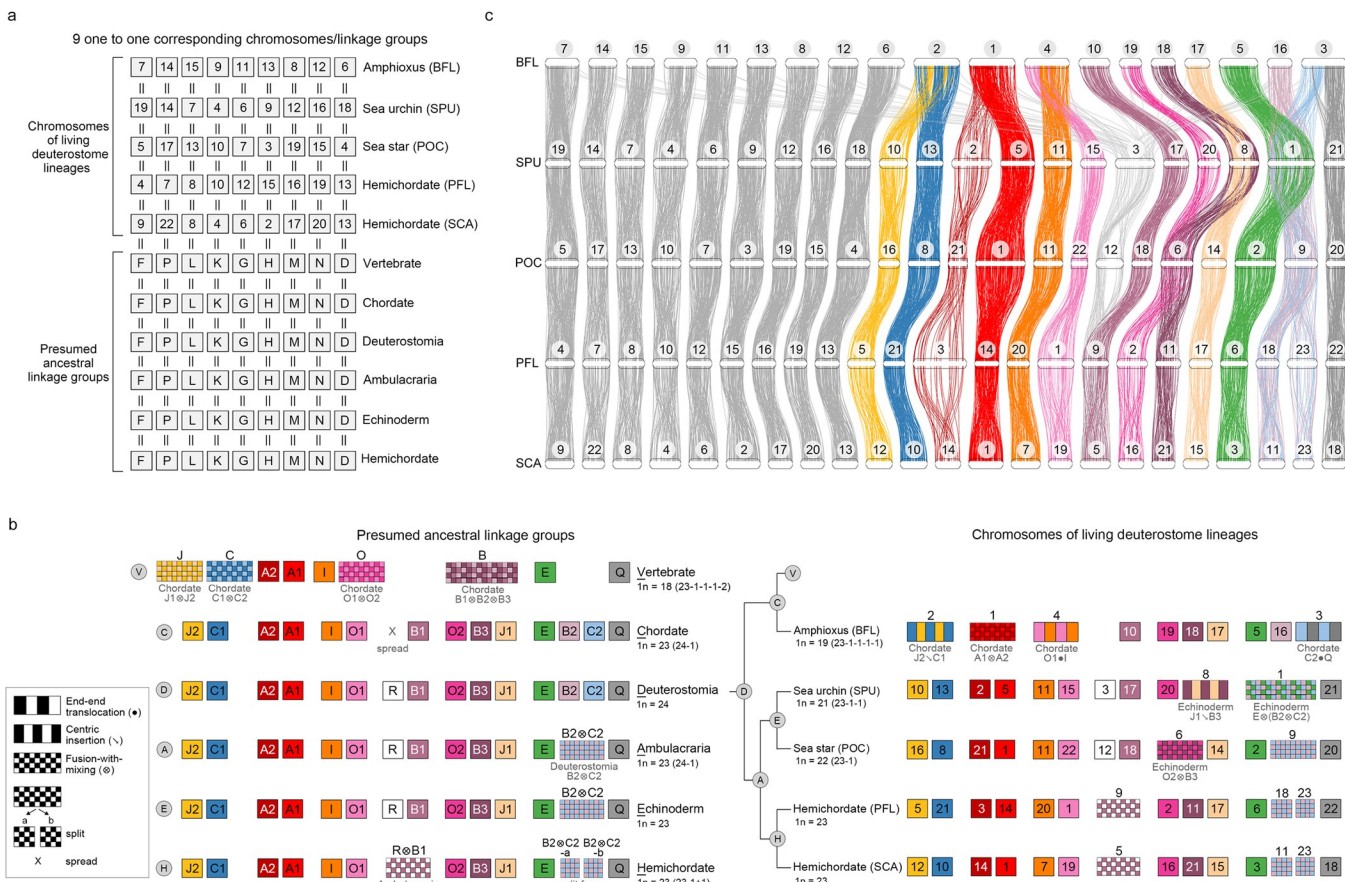

**Fig 2. Evolutionary history of deuterostome chromosome architectures.** (**a**, **b**) A schematic representation of chromosome evolution in deuterostome lineages. The chromosomal architectures of presumed LCAs (bottom in **a** and left in **b**) and the chromosomal architectures of living deuterostome species (top in **a** and right in **b**). Each box denotes an individual chromosome. Haploid number (1N) and increase (+) or decrease (-) in quantity of chromosomes are indicated. The color code of boxes is taken from the previous study on vertebrate ancestral chromosomes, except for the 9 one-to-one corresponding chromosomes (**a**, light gray boxes). Chromosomal architecture of the LCA of vertebrates was based on the previous study [20]. In cases where chromosomal fusion events were deduced, types of changes are indicated below color boxes with symbols defined previously [19]; end-end translocation (●), centric insertion (↘), and fusion-with-mixing (⊗). Box sizes do not reflect the actual sizes of chromosomes. (**c**) Chromosomal positions of the orthologous gene pairs among 5 deuterostome species. Horizontal bars with numbers on top represent chromosomes of each species. In total, 3,668 orthologous gene pairs are illustrated. For ease of comparison, the chromosome sizes are scaled proportionally such that the 5 genome assemblies reach equal sizes. Except for the genes that spread into multiple chromosomes in amphioxus (BFL), gene pairs that are not located on the corresponding chromosomal pairs or cannot be found in all 5 species are not shown. The data underlying this figure can be found in S1 Data. BFL, *Branchiostoma floridae*; LCA, last common ancestor.

therefore likely a shared derived character of the superorder Echinacea, a hypothesis that can be tested by sequencing other members of this group.

In the chordate lineage (node C in Fig 2B), orthologous genes located on DALG R were dispersed into many chromosomes [19], leading to 23 chordate ALGs (CALGs). This dispersion was inferred from the observation that no particular amphioxus chromosomes show significant enrichment of syntenic blocks corresponding to DALG R-derived chromosomes in echinoderms (SPU3 or POC12, Figs 2C, S4E and S4F). Similarly, no concentration of R was found in vertebrates or ascidians [19]. In the amphioxus *B. floridae*, 4 chromosomal fusion events occurred (J2↘C1, A1⊗A2, O1●I, and C2●Q), reducing the number of chromosomes to 1N = 19 [20]. The inferred chordate-specific chromosomal dispersion and the 4 chromosomal fusion events in amphioxus BFL are consistent with previous findings [19]. One of these fusion events (A1⊗A2) was also observed in the sea urchin *Paracentrotus lividus* [26], suggesting that

A1 and A2 were arms of a metacentric chromosome that fused independently in urchin and amphioxus. From the 23 chordate ALGs, previous studies [19,20] deduced that the lineage leading to vertebrates had undergone 4 chromosomal fusion events (J1⊗J2, C1⊗C2, O1⊗O2, and B1⊗B2⊗B3), reducing the 23 CALGs to 18 vertebrate ALGs. These chromosomal rearrangement events and the evolutionary history of genomic architectures among deuterostomes are summarized in Fig 2.

## Stepwise changes in chromosomal architectures within the sea urchin lineage

We expect that chromosomal fusion-with-mixing events would occur in a stepwise process as evolution proceeds. As such, 2 distinct chromosomes (at $t_0$) would fuse (at $t_1$), either by end-end fusion or centric insertion, and this event would be followed by rounds of intrachromosomal inversions and translocations (at $t_2$) until the fused chromosome became scrambled (at $t_s$) (as illustrated in S10 Fig). We therefore postulate that comparing chromosome architectures between species with a relatively short divergence time should allow us to identify the evolutionary state of individual chromosomes during this stepwise process. We thus analyzed 2 additional sea urchin species, *L. variegatus* (LVA) and *L. pictus* (LPI), for which chromosomal-level genome assemblies are available for syntenic comparison [16,27]. LVA and LPI are within the genus *Lytechinus*, which share a common ancestor with *S. purpuratus* 50 million years ago (mya) [28]. By analyzing syntenic conservation of these 3 sea urchin species (S11 Fig), we inferred that their LCA (tentatively assumed to be sea urchin LCA) possessed 21 ALGs (SALGs) due to 2 shared chromosomal fusion events, J1↘B3 and E⊗(B2⊗C2) (node S in S10 Fig). These 2 fusions were also observed in the recently decoded sea urchin *P. lividus* genome [26], indicating a common genomic trait of currently available sea urchin genomes. We also deduced 20 ALGs (LALGs) in the *Lytechinus* LCA, owing to a *Lytechinus*-specific chromosomal fusion event (G●D) (node L in S10 Fig). Descending from the *Lytechinus* LCA, *L. variegatus* and *L. pictus* each underwent a distinct chromosomal fusion event, F●(J1⊗B3) into *L. variegatus* LVA1 and F●C1 into *L. pictus* LPI5, independently resulting in $1N = 19$ chromosomes for both species.

Based on the phylogenetic relationships and deduced chromosomal architectures (S10 Fig), we construct a putative history of several chromosomal fusion events. For example, 2 echinoderm ALGs (J1 and B3 at $t_0$) fused via centric insertion after which a translocation event resulted in the sea urchin ALG J1↘B3 (at $t_2$). This chromosome then underwent extensive recombinations to become the *Lytechinus* ALG J1⊗B3 (at $t_s$). In the lineage leading to *L. variegatus*, but not *L. pictus*, end-end fusion of *Lytechinus* ALGs F and J1⊗B3 resulted in the extant LVA1 chromosome (at $t_1$). Within the LVA1 chromosome, we observed no obvious translocation between regions descended from LALGs J1⊗B3 and F, suggesting that the end-end fusion likely occurred recently in the lineage leading to *L. variegatus*. In *L. pictus*, chromosome LPI5 was derived from end-end fusion of LALGs F and C1 followed by a translocation event. Intriguingly, the independent, species-specific fusion event of the 2 *Lytechinus* species involved the same chromosome (LALG F). Such recent chromosomal fusions may alter recombination rate and cause reproductive isolation, as observed during nematode speciation [29]. Together, the fusion events in sea urchins clearly illustrate how stepwise changes may occur in chromosomal architectures.

In several fusion-with-mixing cases, we did not observe transitional states (e.g., SALG E⊗(B2⊗C2) resulted from EALGs E and B2⊗C2, S10 Fig), implying that these fusion events occurred at a relatively ancient time. Assuming that intrachromosomal rearrangements occurred at a constant rate, we postulate the order of fusion events based on synteny patterns.

For example, in comparison with the centric insertion pattern of SALG J1↘B3, SALG E⊗ (B2⊗C2) exhibits fusion-with-mixing, suggesting that the fusion of EALGs E and B2⊗C2 occurred earlier than that of EALGs J1 and B3. Therefore, from the echinoderm LCA that possessed 23 ALGs to the sea urchin LCA (or more specifically, the LCA of the 3 sea urchin species under investigation) that contained 21 ALGs, there may have been a transitional state with $1N = 22$ chromosomes, when EALGs E and B2⊗C2 were already fused but J1 and B3 remained separated. Intriguingly, it has been reported that the haploid genomes of *Cidaris cidaris* and *Arbacia punctulata*, which respectively belong to an early branching sea urchin group and an euechinoid outgroup of *Lytechinus* and *S. purpuratus*, each contain 22 chromosomes [30,31], suggesting that only 1 fusion event occurred in early branching sea urchins. Thus, we hypothesize that EALGs E and B2⊗C2 fused before the divergence of the sister subclasses of sea urchins, cidaroids, and euechinoids, at least 268 mya [32]. The second fusion event, involving EALGs J1 and B3, possibly occurred later, after the emergence of *Arbacia* and before the divergence of *Lytechinus* and *S. purpuratus* (i.e., between ~185 and 50 mya) [33]. If that is the case, the LCA of all living sea urchins would have possessed $1N = 22$ chromosomes, instead of the presumed 21 ancestral chromosomes illustrated in S10 Fig. Future synteny analyses and chromosomal architecture reconstructions using genomes of early branching sea urchins will help to resolve this question.

## Lineage-specific chromosomal fusion events in major animal groups

To understand whether the deuterostome chromosomal architectures differ from those of protostomes, we extended our analysis to include several recently published chromosome-level genome assemblies of protostomes. Consistent with previous observations [17,25], we found that the chromosomes of most protostome species are highly rearranged. Nevertheless, we were able to identify genomes of 5 spiralian species [25,34–37], including 3 bivalves (2 clam species, *Ruditapes philippinarum* and *Sinonovacula constricta*, and the aforementioned scallop *P. yessoensis*) and 2 polychaete annelids (*Paraescarpia echinospica* and *Streblospio benedicti*), which are more conserved and comparable to the presumed bilaterian ALGs and extant deuterostome genomes. Our syntenic analysis shows that all the 5 spiralian species share 4 specific fusion-with-mixing events (S12 and S13 Figs), as predicted previously based on 4 syntenic synapopmorphies of spiralians identified using different datasets [19,38]. Comparisons of 6 chromosome-scale ecdysozoan genomes, however, showed that they are highly reorganized relative to the bilaterian ancestor [19], making it difficult to reconstruct the chromosomal architecture of their LCA. The 4 spiralian fusions, however, are clearly absent in ecdysozoan, consistent with their status as spiralian syntenic synapomorphies [19]. For example, these 4 fusion events are clearly absent in 2 butterfly genomes (S14 Fig). Based on these pairwise syntenic comparisons, we inferred that the LCA of protostomes most likely also possessed 24 ALGs that correspond to the 24 BALGs (S12 Fig). This correspondence suggests that the genomic architecture of the deuterostome LCA and protostome LCA did not undergo large-scale interchromosomal fusions when they initially diverged from the bilaterian LCA. However, during subsequent evolution, protostome lineages appear to have accumulated much more extensive changes in their chromosomal architectures than deuterostome lineages.

After chromosomal fusion with extensive mixing, it is unlikely that genes in a fused chromosome would be sorted to reassemble back into individual chromosomes with the original makeup [18,19], and, such irreversible chromosomal fusion-with-mixing events can be used as polarized traits for probing deep phylogenetic relationships of animals [18,19]. Recent molecular phylogenomic studies have provided evidence to support the sister group relationship between Ambulacraria and Xenacoelomorpha, and some even questioned the monophyletic

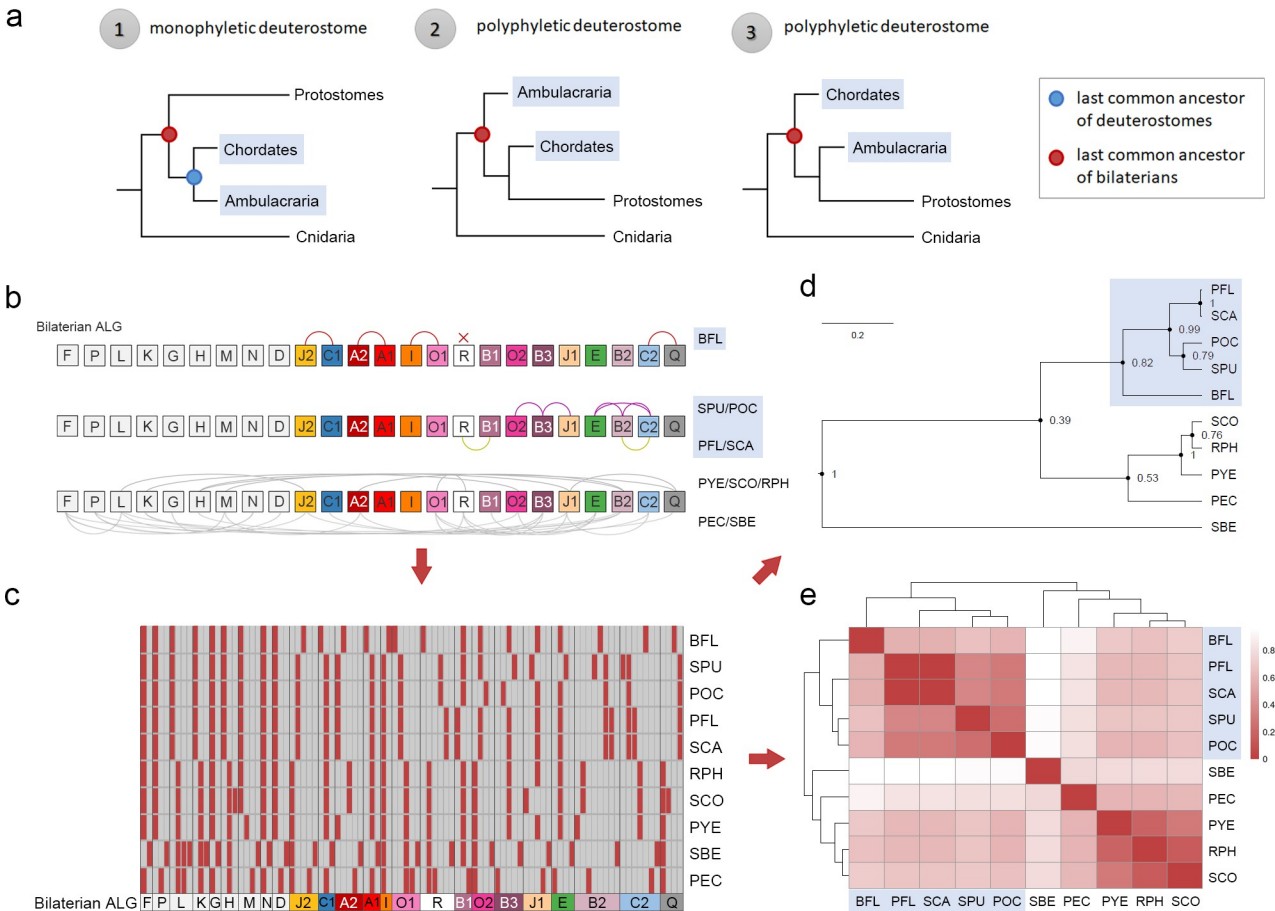

**Fig 3. Category clustering analysis based on rearrangement events of the 24 bilaterian ancestral chromosomes.** (**a**) Three scenarios of phylogenetic relationships among bilaterians can be postulated. In the first scenario (monophyletic deuterostome), 2 deuterostome branches, chordates and ambulacrarians, are grouped together, and their LCA (the LCA of deuterostomes) is denoted with a blue dot. The LCA of bilaterians is indicated with a red dot. In the other 2 scenarios, one of the deuterostome branches is grouped with protostomes, resulting in polyphyletic deuterostomes. In these latter 2 scenarios, the LCA of deuterostomes and bilaterians is the same. (**b**) Distinct chromosomal rearrangement events of each species, including fusion, split, and spread events, are recorded into the category data based on changes deviated from the $1N = 24$ BALGs. For example, there are 3 categories for the bilaterian ALG K, including (1) no rearrangement event (BFL, SPU, POC, SCA, PFL), (2) O2⊗K (RPH, SCO, PYE, PEC), and (3) J1⊗(O2⊗K) (SBE). (**c**) Conversion of the category data into a binary data matrix. Dark vertical lines distinguish different chromosomes. Red box denotes the chromosomal status of each species as compared to the BALGs. For the 10 species that we examined, the number of the chromosomal rearrangement categories ranges from 2 (BALGs F, G, N, and I) to 8 (BALG B2). A detailed binary code table is provided in S1 Data. (**d**, **e**) Bayesian phylogenetic analysis (d) and clustering analysis (e) based on the binary data shows that the 5 deuterostome species (shaded in blue) are grouped together. The data underlying this figure can be found in S1 Data. BALG, bilaterian ancestral linkage group; LCA, last common ancestor.

grouping of Deuterostomia [9–12] (Fig 3A). We asked whether the identified chromosomal fusion-with-mixing traits could help to resolve this issue. We coded chromosomal status into category data, which was then converted into a binary matrix (Fig 3B and 3C). Bayesian phylogenetic and clustering analyses based on these synteny-based characters united the deuterostomes as a clade to the exclusion of other animals (Fig 3D and 3E). Notably, all the 5 deuterostome species we analyzed retain 9 one-to-one matching chromosomes corresponding to the ancestral deuterostome state, however, no common chromosomal fusion (i.e., syntenic synapomorphy [18]) was identified.

Regarding derived chromosomal changes within deuterostomes, we identified an ambulacrarian-specific chromosomal fusion (B2⊗C2) and a chordate-specific chromosomal

dispersion (originated from ALG R). Four spiralian-specific chromosomal fusion events have been described (L⊗J2, O2⊗K, Q⊗H, and O1⊗R) (S16 Fig). We also noted that the bilaterian chromosomal rearrangement events were not observed in the jellyfish (*Rhopilema esculentum*, RES) genome [19] (S15 and S16 Figs). Therefore, the 5 major extant animal groups (i.e., ambulacrarians, chordates, spiralians, ecdysozoans, and cnidarians) do not share common derived traits in terms of inter-chromosomal rearrangement events, and the observed chromosomal fusion events appear to be lineage-specific and have occurred before the diversification of each of these major animal groups.

Xenacoelomorpha, a group comprising xenoturbellids and acoelomorphs, have been placed as either early branching bilaterians (Nephrozoa hypothesis) or as a sister group of ambulacrarians (Xenambulacraria hypothesis) [11,12,39,40]. To test these hypotheses, we examined the recently available chromosome-level genome assembly of the xenoturbellid *Xenoturbella bocki* [41]. We found no evidence of the ambulacrarian-specific chromosomal fusion (B2⊗C2) in the *X. bocki* genome. This fusion event therefore appears to be specific to ambulacrarians and does not provide evidence supporting the Xenambulacraria hypothesis. However, the Xenambulacrarian hypothesis could not be ruled out by the current data, as the fusion could have occurred in the ambulacrarian lineage after Ambulacraria diverged from Xenacoelomorpha. Overall, our results reinforce the idea that the branch length between bilaterian LCA and deuterostome LCA is likely very short [9], and our analyses also show that deuterostome lineages experienced fewer chromosomal fusion events than protostomes during early bilaterian evolution.

## GO enrichment analyses of lineage-specific chromosomal rearrangement events

Chromosomal fusion-with-mixing has the potential to disrupt long-range promoter-enhancer interactions and/or topological association domains (TADs) to cause changes in gene regulation. The genes present on chromosomes that underwent lineage-specific fusions could therefore provide hints as to the origins of lineage-specific novelties. To assess the potential biological consequences of specific chromosomal changes in deuterostome species, we performed gene ontology (GO) enrichment analyses on genes located on the corresponding chromosomes of extant deuterostomes. The ambulacrarian-specific chromosomal fusion-with-mixing resulted in the inferred AALG B2⊗C2, which has remained as a single chromosome POC9 in the sea star (Figs 2B and 4). We found that genes located in POC9 are enriched in several GO terms related to development, including germ layer formation, neural development, axial patterning, gastrulation and regulation of BMP and Wnt signaling pathways (Figs 4A and S17E). This observation suggests that in the lineage leading to ambulacrarians, many developmental regulatory genes would have experienced extensive shuffling in their relative positions via chromosomal fusion-with-mixing (B2⊗C2), which could have altered their expression patterns. The fused AALG B2⊗C2 further underwent distinct chromosomal fusion and splitting events in sea urchins and hemichordates, respectively (Figs 2B and 4).

In all the 3 sea urchin genomes we analyzed, a single chromosome (e.g., SPU1) was derived from the fusion of EALGs E and B2⊗C2 (S10 Fig). GO analysis revealed that genes related to development are also enriched in SPU1 (Figs 4B and S18E). Intriguingly, genes involved in bone and otolith development are also enriched in this sea urchin-specific fusion chromosome. Further analysis on genomes of other sea urchin species and functional experiments will be required to determine whether the rearrangement of these genes is related to the emergence of the unique skeletogenic lineage of sea urchins.

In both hemichordate species, we inferred that 2 chromosomes (PFL18 and PFL23 of *P. flava* and SCA11 and SCA23 of *S. californicum*) were split from the fused AALG B2⊗C2,

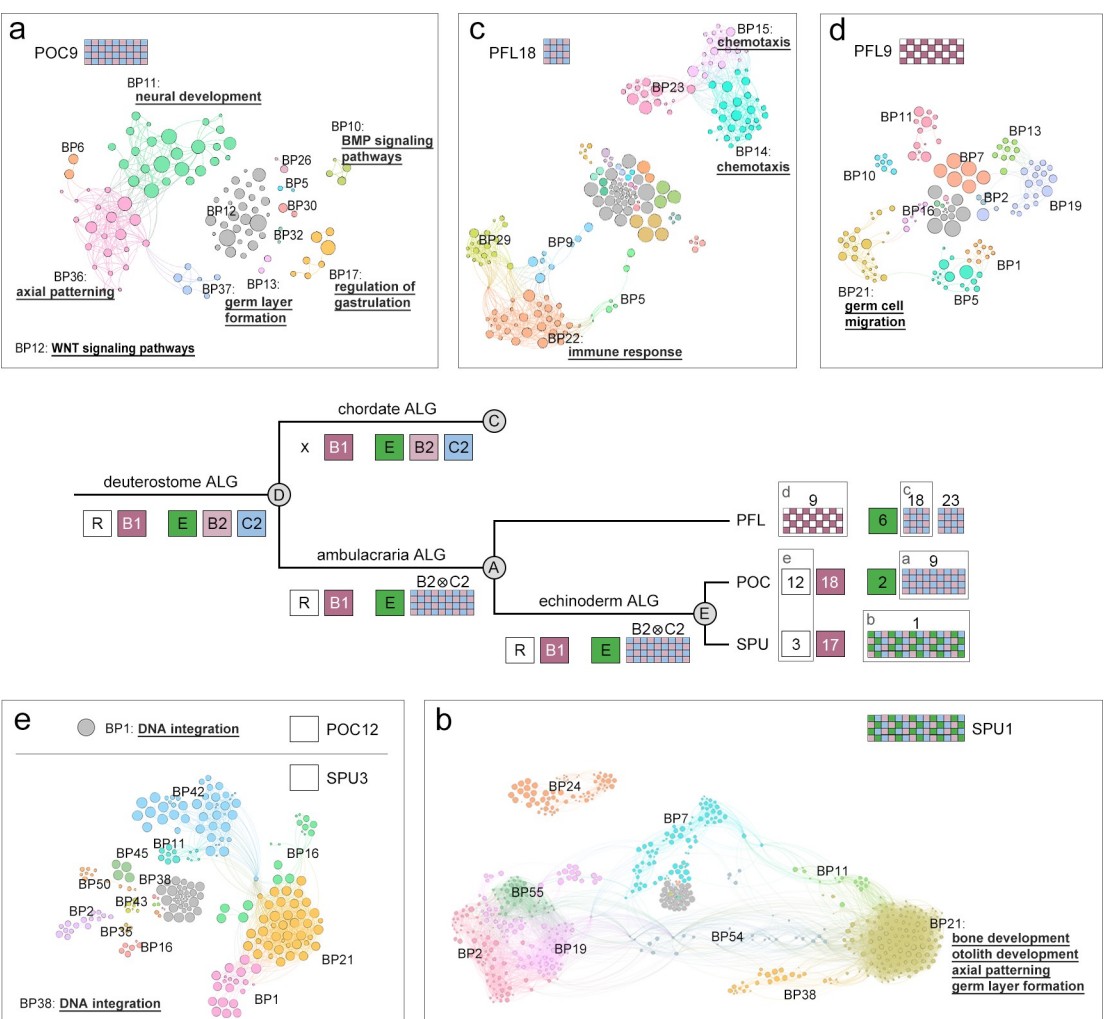

**Fig 4. GO enrichment analyses of chromosomes that underwent lineage-specific changes in deuterostomes.** GO enrichment analysis of genes located in the sea star POC9 (**a**), sea urchin SPU1 (**b**), hemichordate PFL18 (**c**), and PFL9 (**d**). The echinoderm chromosomes (POC12 and SPU3) corresponding to deuterostome DALG R were also analyzed to understand the chordate-specific chromosomal dispersion (**e**). The enriched GO terms (adjusted $p$-value <0.1) are clustered and divided into different modules, and selected terms are underlined. The top 3 enriched BP GO terms of each module are shown in S17–S19 Figs. The data underlying this figure can be found in S2, S3, and S4 Data. BP, biological process; GO, gene ontology.

resulting in HALGs B2⊗C2-a and B2⊗C2-b in the LCA of hemichordates (Fig 2B). GO enrichment analysis revealed that genes located on PFL18 (descendant of either HALG B2⊗C2-a or B2⊗C2-b) are enriched in biological processes associated with immune response and chemotaxis, suggesting that distinct interactions with environmental factors could have emerged during hemichordate evolution via chromosomal rearrangement (Figs 4C and S19C). Additional lineage-specific fusion events observed in deuterostomes include the echinoderm O2⊗B3 and J1↘B3 (resulting in the sea star POC6 and the sea urchin SPU8, respectively) and the hemichordate-specific fusion R⊗B1 (corresponding to PFL9 and SCA5) (Fig 2B). The top GO terms enriched in POC6, SPU8, and PFL9 include neuronal regulation, thyroid hormone transport, and germ cell migration, respectively (Figs 4D and S17–S19).

All chordates appear to share a dispersal of deuterostome/bilaterian ALG R [19], but this ALG is retained as individual chromosomes in ambulacrarians (e.g., POC12 and SPU3)

(Fig 2B and 2C). Intriguingly, we found that POC12 and SPU3 are enriched for genes involved in DNA integration, including several transposase genes (Figs 4E, S17A and S18A). This result suggests that the dispersion of DALG R in the chordate lineage could have been due to the mis-regulation of transposase genes or rearrangements induced by such sequences. Taken together, our GO enrichment analyses provide a global view of possible regulatory and functional changes related to the lineage-specific chromosomal rearrangements. Such rearrangement events are in agreement with levels of divergence in gene expression profiles [42], supporting the hypothesis that at least some of these potential changes are plausibly associated with the evolution of distinct lineage-specific features and diverse body plans in deuterostomes.

## Hox clusters in rearranged chromosomes

Hox genes are typically arranged in clusters and specify bilaterian body regions along the ante-roposterior axis [43]. Contrary to their structural and functional conservation, we find that Hox clusters are located in chromosomes that underwent fusion with extensive mixing among the 10 bilaterian species we examined, with the sole exception of amphioxus BFL16 (S16 Fig). In the LCA of bilaterians, the Hox cluster was inferred to be positioned in BALG B2. The descendant of this ALG (DALG B2) contributed to the ambulacrarian-specific fusion with DALG C2 to form AALG B2⊗C2. Subsequently, its descendant in echinoderms further under-went an additional fusion-with-mixing with ALG E to give rise to a chromosome resembling SPU1 in sea urchins. Meanwhile, in hemichordates, AALG B2⊗C2 split into HALGs B2⊗C2-a and B2⊗C2-b (represented by the extant PFL18 and PFL23, S16 Fig). Intriguingly, this split-ting event in the hemichordate ancestor separated the Hox cluster and the *distalless* gene, which are commonly linked in vertebrate genomes [44]. This genetic feature appears to be unique to hemichordates, as the Hox cluster and *distalless* gene are located in the same chro-mosome in all other deuterostome species we examined (i.e., amphioxus BFL16, sea star POC9, and sea urchin SPU1). Nevertheless, it remains unclear whether the separation of the Hox cluster and *distalless* gene during the hemichordate-specific chromosomal split would have resulted in functional consequences related to the origin of the hemichordate body plan. BALG B2 is also involved in different fusion-with-mixing events in the 5 spiralian species, with the spiralian Hox clusters located on the highly rearranged RPH14, SCO9, PYE1, PEC4, and SBE9 (S16 Fig). It is tempting to speculate that these chromosome rearrangement events may have changed the regulatory landscape of Hox genes and contributed to the evolution of lineage-specific body plans. Further studies would certainly be required to test this hypothesis.

While intrachromosomal rearrangement events are highly associated with the accumula-tion of transposable elements (TEs) [45,46], Hox clusters are known to be largely devoid of TEs in chordates [47,48]. The exclusion of TEs from Hox clusters is thought to be chordate-specific, as this trend was not detected in 5 protostome species that have been analyzed (including 4 insects and the nematode *Caenorhabditis elegans*) [48]. The observation that most Hox clusters are situated in chromosomes that underwent fusion-with-mixing prompted us to analyze TE densities in the Hox-bearing chromosomes. We observed a clear drop-off of TE densities (including DNA transposons (DNA), long terminal repeats (LTR), long interspersed nuclear elements (LINE), and short interspersed nuclear elements (SINE)) within Hox clusters compared with the non-Hox regions of the same chromosomes; this trend was observed in all 9 bilaterian species we examined (Figs 5A and S20–S22). The overall TE densities in Hox-bear-ing chromosomes were similar to the densities observed across entire genomes (S23 Fig). The exclusion of TEs in Hox clusters is particularly apparent in amphioxus BFL and hemichordate PFL (approximately 77% less than the density of non-Hox regions) in which the Hox clusters are relatively intact (Fig 4). Therefore, the trend of lower TE densities in Hox clusters is

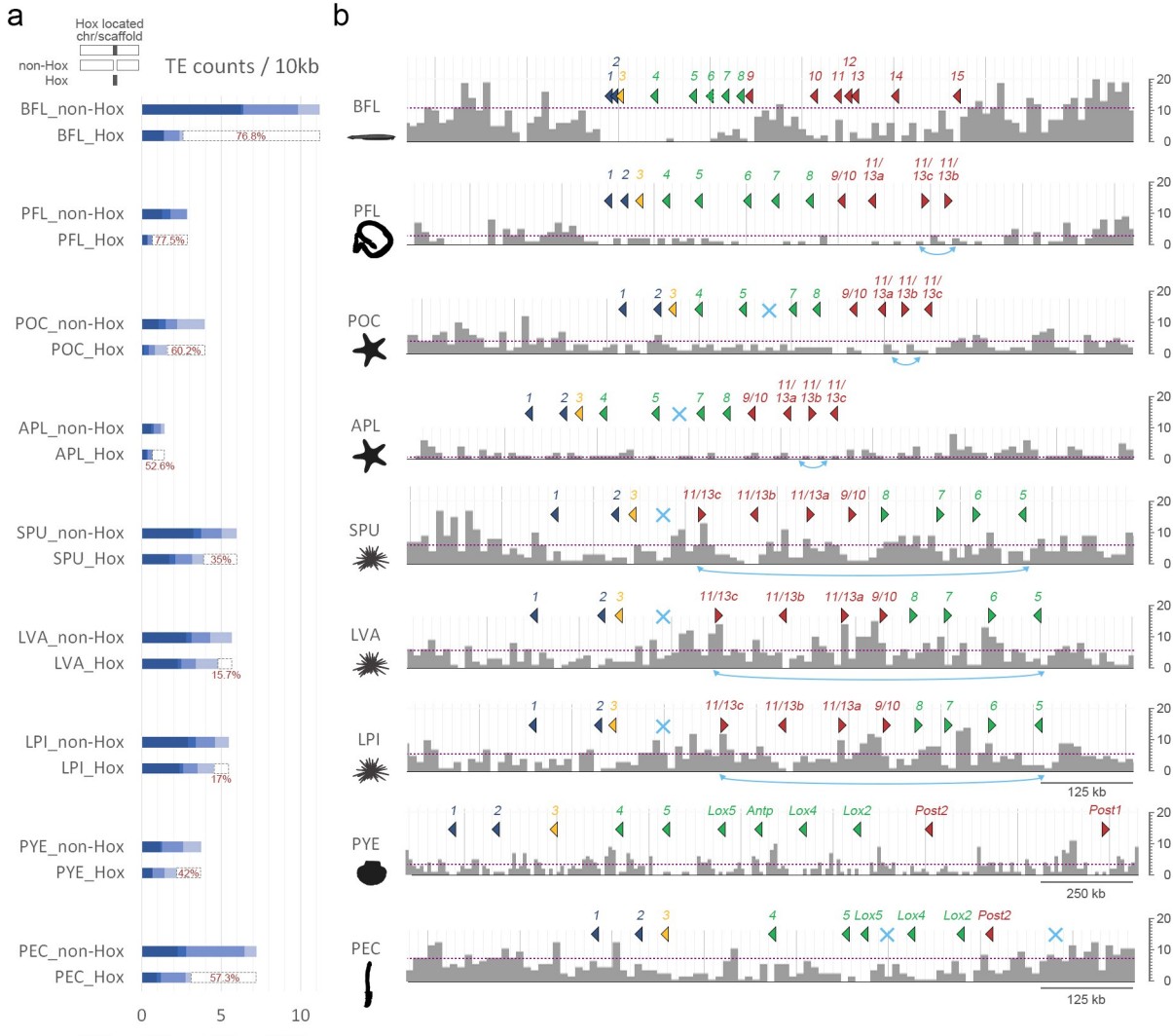

**Fig 5. Counts of TEs around the Hox-bearing genomic regions.** (**a**) Densities of all TEs (DNA + LTR + LINE + SINE) within the Hox cluster region and non-Hox region of the Hox-bearing chromosome/scaffold of each species. The percent differences in normalized TEs counts between the non-Hox region and the Hox cluster region are illustrated (dashed bars and red values). (**b**) Distributions of all TEs around the Hox gene cluster of each species. The bin size for each histogram is 10,000 bp. Dotted lines indicate the averaged TE densities of the Hox-bearing chromosomes/scaffolds. Color coding denotes division of Hox genes in "anterior" (dark blue), "group 3" (yellow), "middle" (green), and "posterior" (red) groups. The light blue crosses represent missing Hox genes. Double arrows in light blue indicate inversion events of Hox genes. The data underlying this figure can be found in S1 Data. LINE, long interspersed nuclear elements; LTR, long terminal repeats; SINE, short interspersed nuclear elements; TE, transposable element.

broadly observed across bilaterians and is not limited to chordates. The mechanism that suppresses TE invasion (either by selection against insertions or inhibition of such mutations) remains in effect even when Hox clusters are situated in otherwise highly rearranged chromosomes.

We also noticed that many genes neighboring Hox clusters, except for the *evx* genes, are highly rearranged and their orthologous genes are commonly found in different chromosomes (S24 and S25 Figs). This result is consistent with the observation that TEs exist at higher densities outside of Hox clusters, where they can promote intrachromosomal rearrangements. Further characterizations of TE distributions within Hox clusters revealed a higher density of TEs

around the posterior Hox genes (between Hox9 and Hox15) within the amphioxus BFL Hox cluster. This higher density is consistent with a previous observation of repeat islands between the amphioxus posterior Hox genes that may contribute to the highly derived posterior region of the amphioxus Hox cluster [47,49,50]. Despite the generally low TE density across the Hox cluster of hemichordate PFL, we noticed that the inversion of *Hox13b* and *Hox13c* coincides with the presence of more TEs near the posterior end of the Hox cluster (Fig 5B, PFL). Similarly, the numbers, positions, and orientations of Hox genes between *Hox5* and *Hox11/13* in the 3 sea urchin species (SPU, LVA, and LPI) have undergone notable changes, which is in line with the higher densities of TEs detected in these regions (Fig 5B).

Taken together, these results indicate that exclusion of TEs from Hox clusters appears to be a conserved feature in bilaterians. Nevertheless, TE invasions sometimes occur in the posterior regions of deuterostome Hox clusters, and these invasions have likely contributed to local rearrangements of Hox genes. Our observations are reminiscent of the proposed "deuterostome posterior flexibility" model, which explains how the posterior Hox genes evolved faster in deuterostomes than in protostomes [50,51]. In conclusion, the distributions of TEs both outside and within certain regions of Hox clusters coincide with intrachromosomal gene rearrangements, which may modify TAD structures of Hox clusters and alter the transcriptional regulation of Hox genes.

## Evolutionary history of the pharyngeal gene cluster

The pharyngeal gene cluster contains 4 transcription factor genes (in the order of *nkx2.1*, *nkx2.2*, *pax1/9*, and *foxa*) and 2 non-transcription factor genes (*slc25a21a* and *mipol1*), and their expression in the pharyngeal slits and surrounding endoderm is considered to be a deuterostome-specific feature [15]. Three additional genes, *msx*, *cnga*, and *egln3*, which respectively encode a homeobox transcription factor, a subunit of cyclic nucleotide-gated channels and Egl-9 family hypoxia inducible factor 3, are also linked to the cluster in some deuterostome species [9,15,52]. The complete pharyngeal gene cluster has so far only been found in deuterostomes, but some of the genes are also linked in protostomes [9]. It has thus been proposed that rather than being a deuterostome-specific trait, an intact cluster may have already been present in the LCA of bilaterians and was later dispersed in protostome lineages [9].

To gain insight into the evolutionary history of the pharyngeal cluster, we analyzed gene complements of the cluster in several bilaterian and non-bilaterian genomes (Fig 6A and 6B). In all the deuterostome genomes we analyzed, we found that *xrn2*, which encodes a 5′ to 3′ exoribonuclease, is associated with the aforementioned pharyngeal genes and usually located upstream and adjacent to *nkx2.1*. Based on the gene repertoire and linkage relationships in the deuterostome genomes, we deduced that the complete complement of the pharyngeal cluster in the LCA of deuterostomes included 10 genes. The complement began with *xrn2*, followed by 3 transcription factor genes (*nkx2.1*, *nkx2.2*, and *msx*), then *cnga*, *pax1/9*, *slc25a21*, *mipol1*, and *foxa*, and finally *egln3*. Several lineage-specific changes then took place within the pharyngeal clusters of deuterostomes (Figs 6B and S26). In the hemichordate PFL, *cnga* was duplicated, and *ghrA* genes invaded the pharyngeal cluster between the *cnga* and *pax1/9* genes. In the sea urchin SPU, the pharyngeal cluster is broken into 3 parts, although the 3 parts are still located on the same chromosome (SPU5), and the second part (including *msx*, *cnga*, *pax1/9*, and *slc25a21*) is inverted.

In all 6 spiralian genomes we analyzed, orthologs of *xrn2* were found to be adjacent to *nkx2.1*, and *mipol1* and *foxa* genes were also linked (Figs 6B and S26). In a previous study [9], paired gene linkages of *nkx2.1* and *nkx2.2*, *pax1/9* and *slc25a21*, and *mipol1* and *foxa* were also identified in various protostomes. These results support the existence of 3 microsyntenic

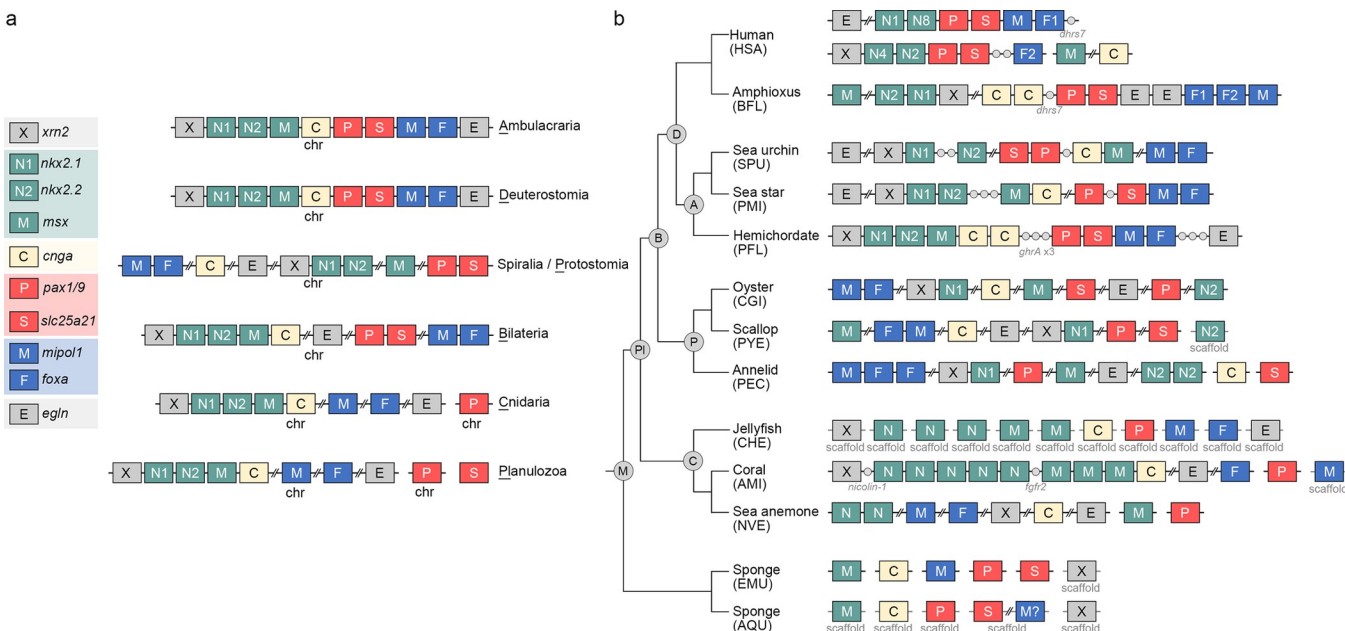

**Fig 6. A possible evolutionary history of the pharyngeal gene cluster.** The pharyngeal gene architectures are shown for presumed last common ancestors at key phylogenetic nodes (**a**) and selected living metazoan species (**b**) (see S26 Fig for the complete dataset). Genes that are commonly linked together are shown in the same color; homeobox-containing genes, including *nkx2.1*, *nkx2.2* and *msx*, are in green, *mipol1* and *foxa* genes are in blue, and *pax1/9* and *slc25a21* are in red. The gray circles indicate genes that are located within the pharyngeal gene cluster. Double slashes are introduced when more than 3 genes are located in between 2 pharyngeal genes. Because *pax* genes of cnidarians and sponges do not show one-to-one correspondence with those of bilaterians, we surveyed the locations of all potential *pax* genes and found that none is linked with the other pharyngeal-related genes in cnidarians and sponges.

blocks, including (1) *xrn2* and *nkx2* genes; (2) *pax1/9* and *slc25a21*; and (3) *mipol1* and *foxa*, as conserved features of bilaterians. Intriguingly, most of the orthologous genes of the pharyngeal cluster are located on the same chromosome, regardless of whether the microsyntenic relationships are maintained.

Based on these observations, we considered 2 scenarios for the evolution of the pharyngeal gene cluster: (1) the LCA of bilaterians (similar to the LCA of deuterostomes) possessed a complete pharyngeal gene cluster that later broke up into 3 microsyntenic blocks in protostomes; (2) the LCA of bilaterians (similar to the LCA of protostomes) had the pharyngeal genes arranged in 3 microsyntenic blocks in the same chromosome that became closely linked to form a compact cluster in deuterostomes. To find evidence supporting or excluding these scenarios, we analyzed the genomic positions of the orthologous genes in outgroups to the bilaterians, including several cnidarians and sponges (S26 Fig). In the coral AMI, we observed a syntenic block containing *xrn2*, *nkx2*, *msx*-related, and *cgna* genes. Other cnidarian species either had preserved parts of this syntenic block (e.g., *xrn2* and *nkx2* are adjacent in the coral XSP; *msx*-related and *cgna* are linked in the sea anemone SCAL) or they lacked the syntenic relationships (S26 Fig). Additionally, *slc25a21* was absent in all 6 cnidarian genomes we analyzed. This gene was likely lost in cnidarians, because an ortholog of *slc25a21* was identified in the sponge genomes. Moreover, except for the *pax* genes, orthologs of the other pharyngeal genes are located on the same chromosome of most cnidarian genomes we analyzed. In the 2 sponge genomes, orthologs of the pharyngeal genes are mostly located on different chromosomes or scaffolds, and no microsyntenic blocks were identified. We can therefore infer using the parsimony principle that one microsyntenic block (composed of *xrn2*, *nkx2*, *msx*-related, and *cgna* genes) was already present in the LCA of bilaterians and cnidarians, and the other pharyngeal genes were located on the same chromosomes but had not yet formed

microsyntenic blocks. The 2 additional microsyntenic pairs (*pax1/9-slc25a21* and *mipol1-foxa*) were established in the bilaterian LCA and persist in extant protostomes and deuterostomes. In the lineage leading to the examined spiralian species, the more ancient syntenic block was likely partially disrupted, with only *xrn2* and *nkx2* genes remaining tightly associated. During the evolution of deuterostomes, the 3 microsyntenic blocks became linked and the *egln* gene was added at the end, forming the complete pharyngeal gene cluster.

Our data therefore support a scenario in which the compact pharyngeal gene cluster of deuterostomes was gradually established from preexisting bilaterian microsyntenic blocks on the deuterostome stem. We cannot, however, rule out the scenario in which individual genes or small blocks distributed along an ancestral chromosome assembled into an ordered cluster in the bilaterian ancestor before breaking into 3 microsyntenic blocks in protostomes. Assembly of the 3 microsyntenic blocks into the deuterostome pharyngeal gene cluster plausibly contributes to the co-regulation of the genes. Indeed, similar temporal expression profiles of the pharyngeal cluster genes are observed among deuterostomes, while orthologs of these genes in protostome and non-bilaterian species display more divergent expression profiles [42]. These results support the idea that clustering of the pharyngeal genes in deuterostomes likely contributes to their co-regulation.

## Conclusions

In this study, we generated chromosome-level genome assemblies for 2 hemichordate species. The hemichordate chromosomes ($1N = 23$) exhibit remarkable chromosome-scale macrosynteny when compared to other deuterostomes, including several echinoderm and chordate species. This high level of conservation allows us to infer that the LCA of deuterostomes possessed 24 ALGs, the same complement as inferred for the bilaterian ancestor [19]. We further deduced lineage-specific chromosomal rearrangement events that resulted in reduced numbers of chromosomes during deuterostome evolution. Genes distributed in chromosomes that underwent lineage-specific fusions are enriched for functions in developmental processes, immune responses and chemotaxis. Changes to the regulatory control of these genes may be related to the evolution of distinct lineage-specific features in deuterostome lineages. One example of this concept is the deeply conserved Hox cluster, which is commonly situated in a chromosome that is highly rearranged. Nevertheless, Hox genes in deuterostomes generally remain tightly linked with the posterior Hox genes showing higher flexibility, consistent with the distribution pattern of TEs within the Hox clusters. Another conserved gene cluster, the deuterostome pharyngeal gene cluster, appears to have been established gradually by combining three pre-assembled microsyntenic blocks present in the LCA of bilaterians. Complete clustering likely contributes to the co-regulation of the pharyngeal genes. In summary, these results showcase how the global view provided by comparative genomics can contribute to our understanding of genome evolution. Moreover, the lineage-specific genomic changes identified herein may help to delineate molecular mechanisms driving the evolution of the diverse body plans of deuterostomes.

## Methods

### Sample preparation and sequencing

High molecular weight (HMW) genomic DNA of *Ptychodera flava* (PFL) was extracted using DNAzol (Thermo Fisher Scientific) from the sperm of a single male individual collected from Penghu Islands, Taiwan. The size of the purified HMW genomic DNA was examined using a pulsed-field gel electrophoresis system (BIO-RAD). The genomic DNA was then sequenced by the Dresden Genome Center using the PacBio platform with 60× coverage. For *Schizocardium*

*californicum* (SCA), HMW DNA was extracted from a ripe male *Schizocardium*. To keep secretion of mucus to a minimum, animals were washed several times and kept in ice cold seawater during the sperm extraction process. Male spermaducts were opened with forceps and sperm was pipetted with a glass pasteur pipette and transferred to an Eppendorf tube. Tubes were spun down and excess seawater was removed before being placed on ice. The DNA extraction protocol was adapted from Stefanik and colleagues [53] with a combination of pouring between Eppendorf tubes instead of pipetting and avoiding any vortexing. The genomic DNA was then sequenced using the PacBio platform with 63× coverage.

## Chromosome-level genome assembly

For PFL, the initial genome assembly was generated using MARVEL assembler [54] with PacBio reads. Purge Haplotigs (version 1.1.0) [55] was used to phase the diploid genome assembly onto the haploid assembly. The phased haploid genome assembly was then scaffolded using HiRISE with a HiC library (Dovetail Genomics). The sequences of the genome assemblies were further curated using Pilon (version 1.23.2) [56] with the Illumina short reads. For SCA, the raw read error correction, read trimming and assembly were performed with the Canu assembler (version 1.5) [57]. Canu was configured to run with a genomeSize parameter set at approximately 1.8 GBp or roughly twice the expected genome size due to high heterozygosity. After assembly, 2 rounds of polishing were performed with the Arrow consensus calling algorithm [58]. The completeness of the polished genome assemblies was evaluated by using BUSCO (version 5.1.2) [59] with the dataset metazoa_odb10, which contains 954 BUSCO gene groups.

## Gene prediction and functional annotation

For *Ptychodera flava*, gene models were predicted using a combination of ab initio gene prediction, homology support, and transcriptome sequencing. First, ab initio gene prediction was conducted using the MAKER2 pipeline [60] (Dovetail Genomics). Second, the protein sequences from other species, including mouse, chick, zebrafish, spotted gar, sea lamprey, amphioxus, ascidian, sea urchin, and sea anemone, were aligned to the PFL genome assembly using GeMoMa (version 1.7) [61]. Third, the Illumina RNA-seq short reads from PFL at 16 stages [42] were mapped using STAR aligner (version 2.7.6a) [62]. The subsequent genome-guided transcript reconstruction was conducted with StringTie (version 2.1.4) [63] and CLASS2 (version 2.1.7) [64]. The transcripts were also assembled de novo using Trinity (version 2.11.0) [65] and then mapped to the genome assembly by minimap2 aligner (version 2.17-r941) [66]. Fourth, the full-length transcripts were generated with PacBio technology (Iso-seq), and IsoSeq3 (version 3.3.0, https://github.com/PacificBiosciences/IsoSeq) was used to cluster the IsoSeq transcripts. LoRDEC (version 0.9) [67] was used to curate the Isoseq transcripts with the Illumina RNA-seq short reads. The polished IsoSeq transcripts were then mapped to genome assembly using minimap2. Gene models based on Iso-seq data were then reconstructed with cDNA_Cupcake (version 9.1.1, https://github.com/Magdoll/cDNA_Cupcake). Finally, the reconstructed transcripts from the different shreds of evidence were merged and filtered by EvidenceModeler (version 1.1.1) [68]. The combined gene models were further updated by PASA (version 2.4.1) [69]. The amino acid sequences were predicted from the transcripts using TransDecoder (version 5.5.0, https://github.com/TransDecoder/TransDecoder). Each amino acid sequence was aligned against NCBI metazoa subset of the nr database using Blast2GO/OmicsBox (version 1.3.11) [70] with blastp-fast for gene description. The GO (gene ontology) term for each gene was annotated using Blast2GO/OmicsBox [70–72].

For *S. californicum*, gene prediction was performed as in Marlétaz and colleagues [26]. Briefly, hints for de novo prediction using Augustus [73] were derived from transcriptome and protein alignments. Particularly, proteins from *S. kowalevskii* were aligned using Exonerate (version 2.2.0) [74]. A custom repeat library was constructed and annotated using Repeatmodeler and subsequently used to mask repeated regions in the *S. californicum* genome using Repeatmasker (v.4.0.7, http://www.repeatmasker.org). We filtered out gene models that extensively overlapped with mobile elements. Isoforms and UTR regions were added using PASA [69] leveraging the alignment of the assembled transcriptome.

## The genomic datasets for other species

The genome assemblies and gene annotation files across metazoans were collected from public domains, including human *Homo sapiens* (HSA), amphioxus *Branchiostoma floridae* (BFL); sea urchins *Strongylocentrotus purpuratus* (SPU), *Lytechinus pictus* (LPI), and *Lytechinus variegatus* (LVA); sea stars *Patiria miniata* (PMI), *Acanthaster planci* (APL), and *Pisaster ochraceus* (POC); scallop *Patinopecten yessoensis* (PYE); clams *Ruditapes philippinarum* (RPH) and *Sinonovacula constricta* (SCO); oyster *Crassostrea gigas* (CGI); annelids *Streblospio benedicti* (SBE) and *Paraescarpia echinospica* (PEC); argus *Erebia aethiops* (EAE) and *Aricia agestis* (AAG); prawn *Penaeus chinensis* (PCH); horseshoe crabs *Tachypleus tridentatus* (TTR) and *Carcinoscorpius rotundicauda* (CRO); nematode *Heterodera glycines* (HGL); corals *Acropora millepora* (AMI) and *Xenia sp.* (XSP); jellyfish *Rhopilema esculentum* (RES), *Sanderia malayensis* (SMA), and *Clytia hemisphaerica* (CHE); sea anemones *Nematostella vectensis* (NVE) and *Scolanthus callimorphus* (SCAL); and sponges *Ephydatia muelleri* (EMU) and *Amphimedon queenslandica* (AQU). S1 Table lists the sources and other information on the genome data used in this study. The Braker2 pipeline (version 2.1.6) [75–81], including GeneMark (version 3.62) [82] and AUGUSTUS (version 3.4.0) [73], was used for gene prediction for genomes lacking gene model annotations.

## Genome comparison

Pairwise syntenic comparisons between species were conducted using MCscan (Python version) of JCVI (version 1.0.9) [83,84]. The jcvi.compara.catalog module with the LAST aligner of MCscan was used to identify orthologous gene pairs between 2 species. The parameter C-score was set to 0.99 for filtering the LAST hit to contain the reciprocal best hit. The minimum number of gene pairs in a cluster was set to 1 without a restricted window size. The synteny dot plots were visualized using jcvi.graphics.dotplot module. Chromosomes used in the syntenic comparison were labeled with an abbreviation of the species names and ordered according to size (BFL, PFL, SCA, SPU, POC, PYE, RPH, SBE, and PEC) or the existing names (LPI, LVA, EAE, AAG, PCH, TTR, CRO, HGL, SCO, and RES).

To assign corresponding chromosome pairs between species, Fisher's exact test with Bonferroni correction in the R software environment (version 3.6.3) was used to calculate the quantitative significance of orthologs located on the chromosome pairs. Risk difference was used to judge significantly higher or lower than others. For example, in S3A Fig, the number of ortholog pairs in PFL1 and SPU15 is 202 (a); in PFL1 and non-SPU15 it is 99 (b); in non-PFL1 and SPU15 it is 45 (c); and in non-PFL1 and non-SPU15 it is 8,528 (d). These 4 numbers were subjected to the Fisher's exact test. The significance levels of all chromosomal pairs were examined; the Bonferroni correction was used for multiple comparisons. Subsequently, the risk difference was calculated as $a/(a+b)−c/(c+d)$. The criterion for corresponding chromosome pairs between 2 species was an adjusted *p*-value smaller than 1E-10 and a risk difference value greater than 0. Adjusted *p*-values between 1E-2 and 1E-10 with positive risk differences were

considered to be small-scale chromosomal rearrangement events and are not presented in figures describing the evolutionary history of chromosomal architectures.

Macrosyntenic conservation analysis on the 4 deuterostome species (BFL, PFL, SCA, and SPU) shown in Fig 1B was visualized using the jcvi.graphics.karyotype module of MCscan. The syntenic block was set to a minimum of 4 gene pairs with a maximum distance of 75 genes between 2 matches.

## Clustering and Bayesian phylogenetic analyses

Distinct chromosomal rearrangement events of the 10 bilaterian species were manually recorded into the category data based on changes deviated from the $1N = 24$ bilaterian ancestral chromosomes (ALGs). The category data was subsequently converted into a binary data matrix (S1 Data) and visualized by using the heatmap.2 function of the gplots R package (version 3.1.3). Notably, most species have only 1 category per ALG. However, in some species, an earlier fusion event was also recorded due to the stepwise process during chromosomal evolution. Taking PEC chromosome 2 as an example, the fusion of Protostome ALGs L and J2 occurred, resulting in Spiralian ALG L⊗J2. Subsequently, Spiralian ALGs L⊗J2 and C2 were further fused leading to PEC chromosome 2. As a result, both categories, L⊗J2 and C2⊗ (L⊗J2), for PEC were recorded as "1." The redundant categories were then removed to avoid double counting before clustering analysis. The distance matrix among the 10 bilaterian species was then calculated based on the binary data matrix using the dist function with the binary method in R. The clustering result was visualized with the pheatmap R package (version 1.0.12). Bayesian phylogenetic analysis was conducted using BEAST (version 1.10.4) [85]. First, the manually converted NEXUS file of binary code matrix (S1 Data) was transformed into an XML file using BEAUti with default parameters. After 10,000 randomly sampled trees were generated using BEAST, the consensus tree was generated using TreeAnnotator with 25% burnin and visualized using FigTree (https://github.com/rambaut/figtree, version 1.4.4).

## GO enrichment analysis

The gene list for each selected chromosome was subjected to GO enrichment analyses using Blast2GO/OmicsBox (version 1.3.11) with an adjusted $p$-value (FDR) of 0.05. The REVIGO algorithm (http://revigo.irb.hr/) [86] was then used to remove redundant GO terms based on the semantics. Finally, the enriched GO terms were clustered and visualized by Gephi (version 0.9.5, https://gephi.org/).

## Hox gene cluster

The genome assemblies and gene model files of bilaterians for Hox gene analysis were downloaded from the public domain (S1 Table). Some misannotated Hox genes were manually curated. Repetitive elements for each species were identified de novo using RepeatModeler (version 2.0.1) [87]. RepeatMasker (version 4.1.2-p1, http://www.repeatmasker.org) was then used for searching and quantifying the identified repeats on each genome assembly, including 4 transposable elements: DNA transposons (DNA), LTR, LINE, and SINE. The numbers of the different transposable elements were calculated with a bin size of 10 kilobases or 50 kilobases using BEDTools (version 2.30.0) [88] and deepTools (version 3.5.1) [89]. The genome sequences and transposable element tracks were subjected to visualization using a local genome browser, JBrowse (version 1.16.10) [90]. The silhouettes were downloaded from PhyloPic (https://www.phylopic.org/).

### Pharyngeal gene cluster

The genome assemblies across metazoans were collected from the public domain (S1 Table). For the genome lacking annotations, the Braker2 (version 2.1.6) pipeline, including GeneMark (version 3.62) and AUGUSTUS (version 3.4.0), was used to predict gene models. Protein sequences of known pharyngeal-related genes were used as query sequences to blast the genome assemblies, and the hits were further confirmed by searching the NCBI nr database.

## Supporting information

**S1 Fig. Chromosome-level genome assemblies of the 2 hemichordates.** Statistical data (left) and treemap (right) of *P. flava* (**a**) and *S. californicum* (**b**) genome assemblies based on PacBio long reads; 27 and 23 larger scaffolds of *P. flava* and *S. californicum* were taken into chromosomal sequences and denoted by blue boxes. The green boxes represent the remaining scaffolds.
(TIF)

**S2 Fig. Further analysis of the HiC dataset on the *P. flava* genome assembly using the HiC-pro pipeline.** (**a**) A Hi-C contact map of *P. flava* genome assembly based on the HiC-pro pipeline [91]. Note that the 3′ end of PFL3-1 interacts with the 5′ end of PFL3-2, and the 5′ end of PFL3-1 interacts with the 3′ end of PFL3-3 (green arrows). The boxed area is magnified to show the chromosomal interactions around the PFL chromosome 23. (**b**) The 3′ end (right side) of PFL23-1 interacts with the 3′ end of PFL23-2 (blue arrow), suggesting that the 2 scaffolds are closely linked at their 3′ ends. These 2 scaffolds also highly interact with several smaller scaffolds (blue arrowheads). Similarly, the 3′ end of PFL3-4 interacts with the 5′ end of PFL3-3 (green arrow). Based on the contact information, PFL3-1 to PFL3-4 were assembled in the order of PFL3-4, PFL3-3, PFL3-1, and PFL3-2. PFL3-2 and PFL3-3 also interact with several smaller scaffolds (green arrowheads). *P. flava* chromosome #3 (PFL3) was thus assembled by joining PFL3-1 to PFL3-4; PFL23 was assembled by joining PFL23-1 and PFL23-2. The data underlying this figure can be found in S5 Data.
(TIF)

**S3 Fig. Syntenic dot plots between *P. flava* and 2 deuterostome species.** Each dot denotes an orthologous gene pair identified between 2 hemichordates SCA and PFL (a) or between sea urchin SPU and hemichordate PFL (b). Chromosomes/scaffolds are separated by gray lines. *P. flava* PFL3-1, PFL3-2, PFL3-3, and PFL3-4 (green boxes) correspond to *S. californicum* SCA14 (a) and *S. purpuratus* SPU2 (b), further supporting the conclusion that PFL3-1 to PFL3-4 constitute the same chromosome. PFL23-1 and PFL23-2 (blue boxes) correspond to SCA23 (a) and SPU1 (b), supporting the conclusion that PFL23-1 and PFL23-2 are from the same chromosome. Notably, comparison of the 2 hemichordate genomes did not show apparent microsynteny conservation, suggesting that large-scale intra-chromosomal rearrangements occurred at least in one of the 2 lineages leading to the 2 hemichordate species. The data underlying this figure can be found in S1 Data.
(TIF)

**S4 Fig. Pairwise syntenic dot plots and significant associations between deuterostome species.** Dot plots (upper panels) showing the chromosomal positions of orthologous gene pairs between 2 species. Statistically corresponding chromosomes are shaded based on significance level in Fisher's exact test and risk difference. In the scatter plots (lower panels), the circle sizes depict the -log 10 adjusted *p*-value, with a maximum of 300 for each plot. Adjusted *p*-values <1E-10, between 1E-5~1E-10 and between 1E-2~1E-5 are marked, respectively, with blue,

yellow, and red. Adjusted *p*-values >1E-2 or risk difference <0 are not shown. For PFL3-1 to PFL3-4 and PFL23-1 to PFL23-2, significance of difference was calculated separately. The data underlying this figure can be found in S1 Data.
(TIF)

**S5 Fig. Pairwise syntenic dot plots and significant associations between genomes of deuterostome species and the scallop (PYE).** Dot plots showing the chromosomal positions of orthologous gene pairs identified between scallop PYE and amphioxus BFL (a), hemichordate PFL (b), sea urchin SPU (c), or sea star POC (d). All symbols are the same as those described in S4 Fig. PYE0 is an unplaced scaffold [25]. The data underlying this figure can be found in S1 Data.
(TIF)

**S6 Fig. Chromosome evolution of deuterostome ALGs J2, C1, A2, A1, I, and O1.** (**a**) Reconstruction of deuterostome ALGs J2, C1, A2, A1, I, and O1 based on pairwise comparisons among amphioxus BFL, hemichordate PFL, sea urchin SPU, sea star POC, and scallop PYE. First, the comparison of POC with SPU showed that POC16, POC8, POC21, POC1, POC11, and POC22 have one-to-one correspondence with SPU10, SPU13, SPU2, SPU5, SPU11, and SPU15, respectively (**b**), suggesting that these 6 chromosomes were already present in their LCA (echinoderm ALGs J2, C1, A2, A1, I, and O1). These 6 chromosomes also have one-to-one correspondence with hemichordate PFL5, PFL21, PFL3, PFL14, PFL20, and PFL1 (**c** and **d**), indicating that the existence of these 6 chromosomes could be traced further back to the ambulacrarian LCA (ambulacraria ALGs J2, C1, A2, A1, I, and O1). Comparisons with the amphioxus BFL genome showed that both POC16/SPU10/PFL5 and POC8/SPU13/PFL21 correspond to a single amphioxus chromosome BFL2 (**e–g**). Similarly, POC21/SPU2/PFL3 and POC1/SPU5/PFL14 correspond to amphioxus BFL1; POC11/SPU11/PFL20 and POC22/SPU15/PFL1 correspond to amphioxus BFL4. To infer the deuterostome ancestral condition, scallop PYE was used as an outgroup. This analysis showed that the 6 ambulacraria chromosomes correspond to 6 distinct PYE chromosomes (PYE4, PYE9, PYE16, PYE5, PYE11, and PYE13, see **h–j**), supporting the conclusion that these 6 chromosomes are ancient and were present in the deuterostome LCA (deuterostome ALGs J2, C1, A2, A1, I, and O1). Accordingly, the 3 amphioxus chromosomes (BFL 2, BFL1, and BFL4) correspond to the aforementioned 6 PYE chromosomes (**k**). Therefore, the amphioxus BFL2, BFL1, and BFL4 were formed from respective fusion events between deuterostome ALGs J2 and C1, ALGs A2 and A1, and ALGs I and O1. These 3 fusion events are likely amphioxus-specific because the 6 deuterostome ALGs correspond to 6 vertebrate ALGs [19,20], which support the notion that these 6 chromosomes remained intact in the LCA of chordates (chordate ALGs J2, C1, A2, A1, I, and O1). The data underlying this figure can be found in S1 Data.
(TIF)

**S7 Fig. Chromosome evolution of deuterostome ALGs R and B1.** (**a**) Reconstruction of deuterostome ALGs R and B1 based on pairwise comparisons. Using the same logic as described for S6 Fig, sea star POC12 and POC18 appear to correspond to sea urchin SPU3 and SPU17, respectively (**b**), supporting the conclusion that their LCA possessed these 2 chromosomes (echinoderm ALGs R and B1). Comparison between hemichordate PFL and echinoderm species revealed that POC12/SPU3 and POC18/SPU17 correspond to a single hemichordate chromosome PFL9 (**c** and **d**). This observation suggests a fusion event occurred in the ambulacraria ancestor leading to PFL9 or a split event leading to POC12/SPU3 and POC18/SPU17. Using amphioxus BFL as an outgroup, the analysis showed that POC18/SPU17 corresponds to BFL10 (**e** and **f**), while amphioxus orthologs of POC12/SPU3 genes spread in the

genome and no single BFL chromosome could be assigned to POC12/SPU3. Another outgroup scallop PYE was then used, revealing that POC12/SPU3 and POC18/SPU17 respectively correspond to PYE13 and PYE12 (**h** and **i**). Based on these comparisons, 3 major inferences can be made: (1) both deuterostome and ambulacraria ancestors possessed the 2 distinct chromosomes (deuterostome/ambulacraria ALGs R and B1); (2) at least in the LCA of hemichordates PFL and SCA, ALGs R and B1 were fused, leading to PFL9/SCA5; (3) in amphioxus, orthologous genes of deuterostome ALG R were dispersed to other chromosomes. Notably, in addition to POC12/SPU3, PYE13 also corresponds to POC22/SPU15, explaining the comparability between PYE13 and the hemichordate PFL1 and amphioxus BFL4 (**j** and **k**) and suggesting a fusion event led to PYE13. Consistent with this idea, the hemichordate PFL9 (fused from ALGs R and B1) corresponds to BFL10 (ALG B1) (**g**). It has been proposed that all chromosomes of vertebrates correspond to amphioxus chromosomes [19,20], suggesting that one ancestral chromosome (ALG R) spread to other chromosomes in the LCA of chordates. The scallop chromosome name was labeled and sorted according to chromosome size. Here, PYE12 is chromosome number 13 and PYE13 is chromosome number 12 in the previous study [25]. The data underlying this figure can be found in S1 Data.
(TIF)

**S8 Fig. Chromosome evolution of deuterostome ALGs O2, B3, and J1.** (**a**) Reconstruction of deuterostome ALGs O2, B3, and J1 based on pairwise comparisons. The sea star POC6 corresponds to sea urchin SPU20 and SPU8 (**b**) and hemichordate PFL2 and PFL11 (**c**); SPU20 and SPU8 also correspond to these 2 PFL chromosomes (**d**), indicating that these 2 chromosomes were present at least in the ambulacrarian and echinoderm LCAs, and POC6 resulted from fusion of the 2 ancestral chromosomes (ALGs O2 and B3). Intriguingly, in addition to POC6, SPU8 also corresponds to POC14, while POC14 corresponds to a single hemichordate chromosome PFL17. Consistently, SPU8 corresponds to PFL11 and PFL17 (**d**), indicating that a single chromosome corresponding to POC14/PFL17 is an ancestral trait (ALG J1), while SPU8 resulted from chromosomal fusion (ALGs B3 and J1). Therefore, it can be inferred that the LCAs of ambulacrarians and echinoderms possessed these 3 ALGs (O2, B3, and J1), which remained as individual chromosomes in hemichordates but underwent different fusion events in different echinoderm lineages. Fusion of ALGs O2 and B3 led to sea star POC6, while fusion of ALGs B3 and J1 resulted in sea urchin SPU8. Consistent with this hypothesis, 3 distinct amphioxus chromosomes BFL19, BFL18, and BFL17 correspond to POC6 and POC14 (**e**); SPU20 and SPU8 (**f**); and PFL2, PFL11, and PFL17 (**g**). This correspondence supports the idea that the presence of the 3 ALGs can be traced back to the LCA of deuterostomes and remained in the chordate LCA. This conclusion is further reinforced by the observation that the scallop genome contains 3 distinct chromosomes (PYE3, PYE19, and PYE18) corresponding to POC6 and POC14 (**h**); SPU20 and SPU8 (**i**); PFL2, PFL11, and PFL17 (**j**); and BFL19, BFL18, and BFL17 (**k**). Additionally, the 3 amphioxus chromosomes BFL19, BFL18, and BFL17 have been shown to correspond to 3 distinct vertebrate chromosomes [19,20], supporting the conclusion that the chordate LCA possessed these 3 chromosomes. The data underlying this figure can be found in S1 Data.
(TIF)

**S9 Fig. Chromosome evolution of deuterostome ALGs E, B2, C2, and Q.** (**a**) Reconstruction of deuterostome ALGs E, B2, C2, and Q based on pairwise comparisons. The sea star POC2 and POC9 correspond to sea urchin SPU1 (**b**). POC2 corresponds to a single hemichordate chromosome PFL6, and POC9 corresponds to PFL18 and PFL23 (**c**). These 3 PFL chromosomes (PFL6, PFL18, and PFL23) also correspond to SPU1 (**d**). This observation suggests that the chromosomes in the sea star (POC2, POC9, and POC20) correspond to those in the LCA

of the 2 echinoderm species, while SPU1 resulted from fusion of the 2 echinoderm ancestral chromosomes (echinoderm ALGs E and B2⊗C2). To infer the ambulacrarian ancestral condition, the amphioxus BFL genome was compared to the ambulacrarian genomes. POC2 and PFL6 correspond to a single amphioxus chromosome BFL5, supporting the conclusion that echinoderm ALG E has a deeper root in the ambulacrarian LCA and deuterostome LCA (ambulacraria/deuterostome ALG E). On the other hand, POC9 and both PFL18 and PFL23 correspond to 2 amphioxus chromosomes, BFL16 and BFL3 (**e–g**). Based on this observation, it may be inferred that POC9 could represent the ambulacraria ancestral chromosome (ambulacraria ALG B2⊗C2), and hemichordate PFL18 and PFL23 resulted from a split of ambulacraria ALG B2⊗C2. Notably, in addition to POC9, amphioxus BFL3 also corresponds to POC20 (**e**). POC20 shows one-to-one correspondence with SPU21 and PFL22 (**b–d**), suggesting that an ancestral chromosome was present at least in the LCA of ambulacrarians (ambulacraria ALG Q) and remained intact in the echinoderm lineage (echinoderm ALG Q). To infer the deuterostome ancestral condition and the evolutionary history of BFL3, the scallop PYE genome was compared to those of the deuterostome genomes (**h–k**). The observation that BFL16 corresponds to a single PYE chromosome (PYE1) supports the idea that the deuterostome LCA possessed this chromosome (deuterostome ALG B2). Additionally, BFL3 corresponds to PYE17 and PYE2. PYE2 also corresponds to BFL13 and 2 one-to-one corresponding chromosomes in ambulacrarian species (POC20/SPU21/PFL22 and POC3/SPU9/PFL15). Therefore, the deuterostome LCA likely possessed ALGs C2 and Q. In the lineage leading to ambulacrarians, deuterostome ALGs B2 and C2 fused and became ambulacraria ALG B2⊗C2. Furthermore, BFL3 also corresponds to 2 vertebrate chromosomes [19,20], so the chordate LCA likely inherited deuterostome ALGs C2 and Q, and these 2 chromosomes then fused specifically in amphioxus to become BFL3. The data underlying this figure can be found in S1 Data.
(TIF)

**S10 Fig. Evolutionary history of sea urchin chromosomal architectures.** A stepwise process of sea urchin chromosomal evolution. We divided the process into 4 time points: $t_0$, $t_1$, $t_2$, and $t_s$ (bottom right panel). At "$t_0$," individual chromosomes have not fused. At "$t_1$," 2 chromosomes are fused by either end-end translocation or centric insertion. At "$t_2$," intra-chromosomal translocations occur, although long stretches of chromosomal regions are still maintained. At "$t_s$," extensive intra-chromosomal rearrangements have occurred, and the fused chromosome becomes scrambled (fusion-with-mixing). We deduced 5 major fusion events that occurred during sea urchin chromosomal evolution, as follows. (1) Echinoderm EALGs E and B2⊗C2 fused and mixed to become sea urchin SALG E⊗(B2⊗C2) ($t_0$ to $t_s$ in green). (2) EALGs B3 and J1 fused via centric insertion, followed by translocation to become SALG J1↘B3($t_0$ to $t_2$ in maroon). (3) A *Lytechinus*-specific fusion event resulted from end-end fusion of SALGs G and D without obvious translocation ($t_0$ to $t_1$ in gray). (4) An LVA-specific fusion event involved *Lytechinus* LALGs F and J1⊗B3 without obvious translocation ($t_0$ to $t_1$ in Navajo white). (5) An LPI-specific fusion resulted from end-end fusion of *Lytechinus* LALGs F1 and C1, followed by an intrachromosomal translocation event ($t_0$ to $t_2$ in blue). Box sizes do not reflect the actual sizes of chromosomes.
(TIF)

**S11 Fig. Pairwise syntenic dot plots among sea urchin lineages.** (**a**) Syntenic analysis of sea urchin LVA and LPI shows remarkable microsynteny conservation (i.e., linear relationships between chromosome pairs). Sea urchin LVA2 corresponds to SPU6 and SPU18, indicating LVA2 was fused from 2 ancestral chromosomes (**b**). Similarly, sea urchin LPI2 also corresponds to SPU6 and SPU18 (**c**), suggesting that this fusion event is a common trait in the

*Lytechnus* genus. Furthermore, LVA1 corresponds to SPU8 and SPU19 (**b**), and LPI5 corresponds to SPU13 and SPU19 (**c**), indicating additional lineage-specific fusion events in sea urchin LVA and LPI. The data underlying this figure can be found in S1 Data.
(TIF)

**S12 Fig. Evolutionary history of protostome chromosomal architectures.** The LCA of protostomes likely retained 24 ALGs (PALGs) that show one-to-one correspondence with the 24 bilaterian ALGs. During protostome evolution, different chromosomal rearrangement events occurred in the spiralian and ecdysozoan lineages. All examined spiralian species, including 3 bivalves and 2 annelids, share 4 fusion events (L⊗J2, O2⊗K, Q⊗H, and O1⊗R), indicating that their LCA (presumably the LCA of spiralians) already possessed the 4 fused ALGs, so the overall number of SpALGs is 20. The LCA of the 3 bivalve species is deduced to have the same complement of ALGs (BiALGs) as the SpALGs, and lineage-specific fusion events are found in the 3 bivalves (see S13 Fig). On the other hand, it can be inferred that the 2 annelids share an additional fusion event (SpALGs C2 and L⊗J2), which brings the number of the annelid ALGs (AnALGs) to 19. Notably, the 4 common fusion events in spiralians were not detected in the ecdysozoan species we examined (red crosses over the fused chromosomes) (see S14 Fig). Weak syntenic conservation between chromosomes of ecdysozoans and other bilaterians suggests that ecdysozoans underwent more complex chromosomal rearrangements. Box sizes do not reflect the actual sizes of chromosomes.
(TIF)

**S13 Fig. Pairwise syntenic dot plots of spiralian chromosomes.** Syntenic analysis showing 4 common fusion events in the spiralian genomes. For example, PYE3 (see S5 Fig), RPH2 (a), SCO6 (b), PEC12 (c), and SBE2 (d) correspond to 2 sea urchin chromosomes SPU4 and SPU20. These 2 sea urchin chromosomes were initially derived from 2 bilaterian ALGs (BALGs K and O2, respectively) and also correspond to 2 different jellyfish chromosomes RPE15 and RPE18 (see S14D Fig), supporting the conclusion that PYE3, RPH2, SCO6, PEC12, and SBE2 were all derived from a fused ancestral chromosome in their LCA. These spiralian species also underwent the following lineage-specific chromosomal rearrangement event(s) (e–h and S5 Fig). (1) Both RPH14 and SCO9 resulted from A2⊗B2 (e, f). (2) SCO5 and SCO17 are either products of a fused (J1●(Q⊗H)) and subsequently split chromosomes, or they are duplicates of the fused chromosome. The latter scenario is less likely because we did not detect significant conservation between SCO5 and SCO17 (i, j). (3) PYE1 was from M⊗B2. (4) SBE1 resulted from F⊗(C2⊗(L⊗J2)) (h). (4) PEC1 (B1⊗E), PEC4 (J1⊗B2), PEC6 (P⊗D), PEC7 (B3⊗(O1⊗R)), and PEC9 (M⊗A2) were each fused from 2 annelid ancestral chromosomes (g). (5) SBE2 (J1⊗(O2⊗K)), SBE3 (G⊗M), SBE4 (P⊗N), SBE6 (E⊗(O1⊗R)), SBE7 (A1⊗B3), SBE8 (D⊗A2), and SBE9 (C1⊗B2) were also each fused from 2 annelid ancestral chromosomes (h). The data underlying this figure can be found in S1 Data.
(TIF)

**S14 Fig. Pairwise syntenic dot plots between chromosomes of ecdysozoan species and sea urchin (SPU).** Pairwise genome comparisons between ecdysozoans and sea urchin SPU showing complex chromosomal rearrangement events in ecdysozoan species, including nematode (a), prawn (b), and horseshoe crabs (c and d). The butterfly genome seems more conserved than the other examined ecdysozoans (e and f). The 4 spiralian fusion events were not found in butterflies, as sea urchin chromosomes corresponding to fused spiralian chromosomes match to different butterfly chromosomes (indicated by columns of the same color). The data underlying this figure can be found in S1 Data.
(TIF)

**S15 Fig. Pairwise syntenic dot plots between chromosomes of jellyfish (RES) and bilaterian species.** The identified chromosomal rearrangement events in bilaterians are not found in the jellyfish genome. SPU3 was derived from DALG R, which dispersed into other chromosomes in chordates. Since SPU3 corresponds to RES19, the chordate dispersal event did not occur in the jellyfish (a). BFL3 and BFL16 correspond to different RES chromosomes, while their ALGs (DALGs B2 and C2) fused into ambulacraria AALG B2⊗C2. Thus, the ambulacrarian fusion event was not found in the jellyfish (b). Similarly, PYE1 and PYE17 both correspond to ambulacraria AALG B2⊗C2 and match to different RES chromosomes (c). The 4 shared fusion events in spiralians were not found in the jellyfish genome, as sea urchin chromosomes corresponding to fused spiralian chromosomes match to different jellyfish chromosomes (indicated by columns of the same color) (d). The data underlying this figure can be found in S1 Data.
(TIF)

**S16 Fig. Summary of the identified chromosomal rearrangement events.** The genomic architectures of bilaterians and the outgroup jellyfish RES are illustrated. The chromosomal rearrangement events of the jellyfish RES are depicted based on the color codes of the 24 bilaterian ALGs. Red arrowheads indicate Hox cluster-containing chromosomes. Box sizes do not reflect the actual sizes of chromosomes.
(TIF)

**S17 Fig. Gene ontology (GO) enrichment analyses of the sea star POC chromosomes 12, 6, and 9.** GO enrichment analyses of genes located on the specific chromosomes of the sea star POC. The enriched GO terms (adjusted $p$-value $<0.05$) are clustered and divided into different modules. Descriptions of the most enriched GO terms of biological process (BP) within each module for genes located on POC12 (**a**), POC6 (**c**), and POC9 (**e**). The bars indicate -log10 adjusted $p$-values for the corresponding GO terms. The full list of enriched GO terms, including BP (biological process), CC (cellular component), and MF (molecular function), is provided in S2 Data. Results of the GO enrichment network analysis of genes located on POC12 (**b**), POC6 (**d**), and POC9 (**f**). Each individual node of the network denotes a specific enriched GO term. Different colors represent different modules of GO terms. Unclassified GO terms are labeled in gray color. Sizes of the circles indicate numbers of genes in each GO term. Manually selected GO terms are indicated with asterisks (*).The data underlying this figure can be found in S2 Data.
(TIF)

**S18 Fig. GO enrichment analyses of the sea urchin SPU chromosomes 3, 8, and 1.** GO enrichment analyses of genes located on the specific chromosomes of the sea urchin SPU. The enriched GO terms (adjusted $p$-value $<0.05$) are clustered and divided into different modules. Descriptions of the most enriched GO terms of biological process (BP) within each module for genes located on SPU3 (**a**), SPU8 (**c**), and SPU1 (**e**). The full list of enriched GO terms is provided in S3 Data. Results of the GO enrichment network analysis of genes located on SPU3 (**b**), SPU8 (**d**), and SPU1 (**f**). All labels are consistent with S17 Fig. The data underlying this figure can be found in S3 Data.
(TIF)

**S19 Fig. GO enrichment analyses of the hemichordate PFL chromosomes 9, 18, and 23.** GO enrichment analyses of genes located on the specific chromosomes of the hemichordate PFL. The enriched GO terms (adjusted $p$-value $<0.05$) are clustered and divided into different modules. Descriptions of the most enriched GO terms of biological process (BP) within each module for genes located on PFL 9 (**a**), PFL18 (**c**), and PFL23 (**e**). The full list of enriched GO terms is provided in S4 Data. Results of the GO enrichment network analysis of genes located

on PFL9 (**b**), PFL18 (**d**), and PFL23 (**f**). All labels are consistent with S17 Fig. The data underlying this figure can be found in S4 Data.
(TIF)

**S20 Fig. Distributions of TEs in amphioxus (BFL) and hemichordate (PFL) Hox-bearing chromosomes.** The genome browser screenshots of the Hox-located chromosomes of BFL (a) and PFL (b). Histograms of all TEs (red), DNA transposons (DNA, yellow), long terminal repeats (LTR, green), long interspersed nuclear elements (LINE, blue), and short interspersed nuclear elements (SINE, purple) are shown. The bin size for each histogram of TEs is 50,000 bp or 10,000 bp (indicated on the left). Red boxes denote the genomic regions of the Hox clusters.
(TIF)

**S21 Fig. Distributions of TEs in sea star (POC) and sea urchin (SPU) Hox-bearing chromosomes.** Positions of various types of TEs in the Hox-bearing chromosomes of POC (a) and SPU (b). All labels are consistent with S20 Fig.
(TIF)

**S22 Fig. Distributions of TEs in scallop (PYE) and annelid (PEC) Hox-bearing chromosomes.** Positions of TEs in the Hox-bearing chromosomes of PYE (a) and PEC (b). All labels are consistent with S20 Fig.
(TIF)

**S23 Fig. TE counts in the 10 bilaterian species.** Numbers of all TEs (DNA + LTR + LINE + SINE) in the whole genome assembly and the Hox-bearing chromosome/scaffold of each species. The TE counts were normalized to a fixed genomic distance (10,000 bp). The data underlying this figure can be found in S1 Data.
(TIF)

**S24 Fig. Genes neighboring Hox clusters are highly rearranged.** Positional analysis around Hox clusters based on unidirectional BLAST. The query species is shown in the middle of each panel. The curved lines connect gene pairs of the BLAST best hits. Hox genes are labeled in gray. Up to 20 neighboring genes of anterior and posterior Hox genes are shown and labeled in blue and red, respectively. Orthologous genes that are not located in chromosomes descended from DALGs E, B2, and C2 are omitted. The full list of BLAST comparisons is provided in S6 Data. The data underlying this figure can be found in S1 Data.
(TIF)

**S25 Fig. Positions of Hox-neighboring genes in deuterostomes.** Comparing SPU (**a**), POC (**b**), PFL (**c**), and BFL (**d**) protein query to protein databases of other deuterostomes using blastp around HOX gene clusters. Each panel is a screenshot from S6 Data. The results are sorted by chromosome number followed by the position of the query IDs.
(TIF)

**S26 Fig. Evolutionary history of the pharyngeal gene cluster with the full dataset.** All symbols are consistent with Fig 6.
(TIF)

**S1 Table. List of collected genome assemblies from the public domain.**
(XLSX)

**S1 Data. Data underlying Figs 1B, 2C, 3C, 3D, 3E, 4A, 4B, S3, S4, S5, S6, S7, S8, S9, S11, S13, S14, S15, S23 and S24.**
(XLSX)

**S2 Data. Data underlying Figs 5A, 5E and S17.**
(XLSX)

**S3 Data. Data underlying Figs 5B, 5E and S18.**
(XLSX)

**S4 Data. Data underlying Figs 5C, 5D and S19.**
(XLSX)

**S5 Data. Data underlying S2 Fig.**
(ZIP)

**S6 Data. Data underlying S25 Fig.**
(XLSX)

**S7 Data. A custom Python script for calculating Fisher's exact test.**
(ZIP)

## Acknowledgments

The authors wish to thank the staff at the core facility of the Institute of Cellular and Organismic Biology, and NGS Genomics core facility of the Biodiversity Research Center, Academia Sinica for technical assistance. We appreciate the valuable discussions with Dr. Mei-Yeh Lu. We also thank Marcus Calkins for English editing. We thank Dr. Sanjit Singh Batra for assistance with the *S. californicum* genome assembly.

## Author Contributions

**Conceptualization:** Che-Yi Lin, Christopher J. Lowe, Daniel S. Rokhsar, Jr-Kai Yu, Yi-Hsien Su.

**Data curation:** Che-Yi Lin.

**Formal analysis:** Che-Yi Lin, Ferdinand Marlétaz, Alberto Pérez-Posada, Pedro Manuel Martínez-García, José Luis Gómez Skarmeta, Juan J. Tena, Christopher J. Lowe, Daniel S. Rokhsar, Jr-Kai Yu, Yi-Hsien Su.

**Funding acquisition:** Juan J. Tena, Christopher J. Lowe, Daniel S. Rokhsar, Jr-Kai Yu, Yi-Hsien Su.

**Investigation:** Che-Yi Lin.

**Methodology:** Che-Yi Lin, Siegfried Schloissnig, Paul Peluso, Greg T. Conception, Paul Bump, David R. Rank.

**Project administration:** Yi-Hsien Su.

**Resources:** Paul Peluso, Paul Bump, Yi-Chih Chen, Cindy Chou, Ching-Yi Lin, Tzu-Pei Fan, Chang-Tai Tsai, David R. Rank.

**Validation:** Che-Yi Lin.

**Visualization:** Che-Yi Lin.

**Writing – original draft:** Che-Yi Lin, Jr-Kai Yu, Yi-Hsien Su.

**Writing – review & editing:** Che-Yi Lin, Ferdinand Marlétaz, Juan J. Tena, Christopher J. Lowe, Daniel S. Rokhsar, Jr-Kai Yu, Yi-Hsien Su.

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
