## [Editor Report · Decision Letter 0]

12 Feb 2024

Dear Yi-Hsien, 

Thank you for submitting your manuscript entitled "Chromosome-level genome assemblies of two hemichordates provide new insights into deuterostome origin and chromosome evolution" for consideration as a Research Article by PLOS Biology.

Your manuscript has now been evaluated by the PLOS Biology editorial staff, as well as by an academic editor with relevant expertise, and I'm writing to let you know that we would like to send your submission out for external peer review.

Once your full submission is complete, your paper will undergo a series of checks in preparation for peer review. After your manuscript has passed the checks it will be sent out for review. To provide the metadata for your submission, please Login to Editorial Manager (https://www.editorialmanager.com/pbiology) within two working days, i.e. by Feb 14 2024 11:59PM.

Kind regards,

Roli

Roland Roberts, PhD

Senior Editor

PLOS Biology

rroberts@plos.org

---

## [Decision Letter · Decision Letter 1]

18 Apr 2024

Dear Yi-Hsien,

Thank you for your patience while your manuscript "Chromosome-level genome assemblies of two hemichordates provide new insights into deuterostome origin and chromosome evolution" was peer-reviewed at PLOS Biology. It has now been evaluated by the PLOS Biology editors, an Academic Editor with relevant expertise, and by three independent reviewers. 

Based on the reviews and our Academic Editor's assessment, we are likely to accept this manuscript for publication, provided you satisfactorily address the points raised by the reviewers and the following data and other policy-related requests.

IMPORTANT - Please attend to the following:

a) Please address the concerns raised by the reviewers.

b) Please address my Data Policy requests below; specifically, we need you to supply the numerical values underlying Figs 1B, 2C, 3CDE, 4ABCDE, 5AB, S2AB, S3AB, S4-S9, S11, S13-15, S17DF, S18BDF, S19BDF, S23, S24, either as a supplementary data file or as a permanent DOI’d deposition. I note that you already have an 5 supplementary data files, but these are small and their relationship to the Figure panels is unclear.

c) Please cite the location of the data clearly in all relevant main and supplementary Figure legends, e.g. “The data underlying this Figure can be found in S1 Data” or “The data underlying this Figure can be found in https://zenodo.org/records/XXXXXXXX

d) Please make any custom code available, either as a supplementary file or as part of your data deposition.

We expect to receive your revised manuscript within two weeks. 

*Published Peer Review History*

*Press*

Sincerely,

Roli

Roland Roberts, PhD

Senior Editor

rroberts@plos.org

PLOS Biology

DATA POLICY:

Regardless of the method selected, please ensure that you provide the individual numerical values that underlie the summary data displayed in the following figure panels as they are essential for readers to assess your analysis and to reproduce it: Figs 1B, 2C, 3CDE, 4ABCDE, 5AB, S2AB, S3AB, S4-S9, S11, S13-15, S17DF, S18BDF, S19BDF, S23, S24. NOTE: the numerical data provided should include all replicates AND the way in which the plotted mean and errors were derived (it should not present only the mean/average values).

CODE POLICY

Per journal policy, if you have generated any custom code during the curse of this investigation, please make it available without restrictions upon publication. Please ensure that the code is sufficiently well documented and reusable, and that your Data Statement in the Editorial Manager submission system accurately describes where your code can be found.

DATA NOT SHOWN?

REVIEWERS' COMMENTS:

Reviewer #1:

This is an extraordinary paper that does much to resolve a variety of important open questions, provides exciting new insight, is sure to be a touchstone for many other projects in years to come. It is rigorously executed and well presented.

My comments address a few presentation issues. My relatively short review reflects how well structured and compelling the manuscript is in its current form.

line 108 - give a better indication when introducing these species of how closely related they are within Hemichordata. For example, do they bracket the ancestral node of the group? Their phylogenetic proximity has a big impact on interpreting the comparative results.

line 349 - 346 - This part is confusing. It is odd to have bayesian phylogenetic support for deuterostomes in analysis of a dataset that does not have a synapomorphy for the group. Consistent with this, the posterior support values in Fig. 3D are quite low. I encourage the authors to clarify the presentations of this result - it is one of the questions of widest interest that is addressed here.

line 359 - missing an "and"

line 363-367 - wording is confusing here. Presumably they do share some rearrangement events that preceded their most recent common ancestor. Reword.

line 422 - reword. Current wording makes its sound like the dispersal is not retained.

line 436 - reword to make it clear whether the hox cluster or chromosome is devoid of transposable elements

Reviewer #2:

[identifies herself as Billie J Swalla]

This paper discusses data comparing two hemichordate Enteropneusta (worm-like) genomes, and the importance of these results to chordate origins, although the "new insights" referenced in the title are only new when key references are left out, as is done in this manuscript. Publication is not acceptable unless the early studies that suggested these results are added to the manuscript and references.

This manuscript shows that chromosomal assembly of two enteropneust hemichordate genomes show remarkable synteny - each hemichordate species has 23 chromosomes and synteny analyses with other phyla reveal that the Deuterostome ancestor was likely to have 24 chromosomes. This data is novel and interesting, and worthy of being published, but the background and discussion of ideas of chordate origins leaves out many key contributions. General discussions are presented and specific comments are written out below.

The authors seemed to have missed key early phylogenies that described the Deuterostomes in the context of chordate origins and the ancestor of the Deuterstomes hypothesized to contain gill slits (Cameron et al 2000). Please add this reference to the first paragraph in the Introduction, as the results were inferred from the Deuterostome phylogeny that showed echinoderms and hemichordates as sister groups. This result was disputed at the time, but genomic evidence has continued to agree with this result.

Line 374-375. "This fusion event therefore appears to be specific to ambulacrarians and does not provide evidence supporting either hypothesis." This is a very disappointing and misleading statement, the authors don't seem to believe their own results! If the Xenacoelomorpha do not share the ambulacrarian-specific chromosomal fusion, then it suggests that they are not ambulacraria, or deuterostomes. 

Specific Comments:

1. Abstract and Introduction - "Deuterostomes are a "superphylum". Modern phylogenetics no longer refer to the old Linnaeus way of classifying animals. 

Deuterostomes are a monophyletic group of animals" is a much better way of describing this group of animals.

2. Figure 1 Legend - Ptychodera flava (PFL) and Schizocardium californicum (SCA) should be switched in the Figure legend to match their position in the Figure: Branchiostoma floridae (BFL), Schizocardium californicum (SCA), Ptychodera flava (PFL), and Strongylocentrotus purpuratus (SPU).

3. The colors in the little phylogeny to the left on Figure 1 are confusing, as they do not correspond to the colors on the right. It would be better to leave them black.

References:

Cameron CB, Garey JR, Swalla BJ. 2000. Evolution of the chordate body plan: New insights from phylogenetic analyses of deuterostome phyla. Proc. Natl. Acad. Sci. 97: 4469-4474.

Reviewer #3:

Lin and collaborators present here the results of a study that generated chromosome-level genome assemblies for two hemichordate species: Ptychodera flava and Schizocardium californicum. The authors used comparative genomic approaches to infer the chromosomal architecture of the deuterostome common ancestor and delineate lineage-specific chromosomal modifications. They found that hemichordate chromosomes exhibit remarkable chromosome-scale macrosynteny when compared to other deuterostomes, and can be derived from 24 deuterostome ancestral linkage groups. The study also identified lineage-specific chromosomal fusion events and analysed the potential biological consequences of these rearrangements. Additionally, the authors investigated the evolutionary history of the pharyngeal gene cluster and the distribution of transposable elements within Hox clusters. Overall, the study provides very interesting insights into the evolution of deuterostome genomes and produce some new hypothesis generating ideas: on the "posterior Hox flexibility" depending on transposable elements; and the pharynx having evolving by linkage of pre-existing bilaterian microsyntenic blocks on the deuterostome stem followed by lineage-specific changes.

The manuscript is very well written and structured, and the methods are sounds and properly justified.

I only have minor questions, mainly of it out of curiosity:

1. The GO enrichment analysis on heavily rearranged chromosomes is very interesting. However, do the authors know if this enrichment differs from that observed in other chromosomes? In other words, is it specifically different rather than just stochastically different? Additionally, are these GO terms exclusive to the heavily rearranged chromosomes within each species, or are they also found in the conserved chromosomes?

2. In the GO enrichment experiment, do the authors have any way to correct for the differing (larger vs smaller) number of genes in each chromosome? Is a larger number of genes, for example, related to more GO terms? If so, how should we interpret the results here?

3. The authors mention that lineage-specific changes in the pharyngeal cluster might have contributed to the diversity of pharyngeal structures in deuterostomes. Are they referring to the gain/loss of specific genes within the cluster in different species? If so, could the authors add this information to the manuscript, complementing Figure 6?

4. Typo in Figure 2: Change "chrodate" to "chordate."

---

## [Editor Report · Decision Letter 2]

3 May 2024

Dear Yi-Hsien,

Thank you for the submission of your revised Research Article "Chromosome-level genome assemblies of two hemichordates provide new insights into deuterostome origin and chromosome evolution" for publication in PLOS Biology. On behalf of my colleagues and the Academic Editor, Chris Jiggins, I'm pleased to say that we can in principle accept your manuscript for publication, provided you address any remaining formatting and reporting issues. These will be detailed in an email you should receive within 2-3 business days from our colleagues in the journal operations team; no action is required from you until then. Please note that we will not be able to formally accept your manuscript and schedule it for publication until you have completed any requested changes.

Sincerely, 

Roli

Senior Editor

PLOS Biology

rroberts@plos.org